# Enhanced antibiotic resistance development from fluoroquinolone persisters after a single exposure to antibiotic

Theresa C. Barrett[1,2], Wendy W.K. Mok[3,4], Allison M. Murawski[1,2] & Mark P. Brynildsen[1,3]

Bacterial persisters are able to tolerate high levels of antibiotics and give rise to new populations. Persister tolerance is generally attributed to minimally active cellular processes that prevent antibiotic-induced damage, which has led to the supposition that persister offspring give rise to antibiotic-resistant mutants at comparable rates to normal cells. Using time-lapse microscopy to monitor *Escherichia coli* populations following ofloxacin treatment, we find that persisters filament extensively and induce impressive SOS responses before returning to a normal appearance. Further, populations derived from fluoroquinolone persisters contain significantly greater quantities of antibiotic-resistant mutants than those from untreated controls. We confirm that resistance is heritable and that the enhancement requires RecA, SOS induction, an opportunity to recover from treatment, and the involvement of error-prone DNA polymerase V (UmuDC). These findings show that fluoroquinolones damage DNA in persisters and that the ensuing SOS response accelerates the development of antibiotic resistance from these survivors.

[1] Department of Molecular Biology, Princeton University, Princeton, NJ 08544, USA. [2] Rutgers Robert Wood Johnson Medical School, Piscataway, NJ 08854, USA. [3] Department of Chemical and Biological Engineering, Princeton University, Princeton, NJ 08544, USA. [4]Present address: Department of Molecular Biology and Biophysics, UConn Health, Farmington, CT 06032-3305, USA. These authors contributed equally: Wendy W. K. Mok, Allison M. Murawski. Correspondence and requests for materials should be addressed to M.P.B. (email: mbrynild@princeton.edu)

Persistence is a type of antibiotic tolerance that refers to specific subpopulations of cells with enhanced abilities to survive antibiotic stress[1–4]. Persisters are not mutants, but rather bacteria that find themselves in a favorable phenotypic niche, where they have abundances of DNA, RNA, proteins, and/ or other cellular components that increase their tolerance to specific antibiotics. Importantly, persistence is measured from log-linear plots of survival as a function of time where biphasic killing kinetics are observed. The first death rate corresponds to the susceptibility of normal cells, whereas the second death rate, which is quantitatively lower than the first yet need not be zero, reflects the susceptibility of persisters. In addition, persistence is heterogeneous, with different subpopulations exhibiting tolerance to different antibiotics, and only some demonstrating multidrug tolerance[5–8]. Due to this specialization, it is reasonable to assume that mechanisms and physiology for one type of persister (e.g., persisters to β-lactams) may not hold true for others (e.g., persisters to fluoroquinolones [FQs]).

Arguably, persisters tolerant to β-lactams are the best-studied. A landmark investigation using time-lapse microscopy demonstrated that β-lactam persisters can arise from growth-inhibited bacteria within growing populations[9]. Such results have been confirmed with additional techniques, such as fluorescence activated cell sorting (FACS), which was used to quantify that ~80% of persisters in an exponentially growing E. coli culture arose from the non-growing subpopulation, whereas the remainder originated from cells that were growing prior to treatment[10,11]. Given that β-lactams act by corrupting enzymes involved in cell wall biogenesis (penicillin-binding proteins, PBPs), it is understandable as to why most persisters to this drug class would be from non-growing subpopulations, which have severely limited levels of peptidoglycan biosynthesis[12]. However, not all persister types fit this mold. For example, Wakamoto and colleagues[13] used time-lapse microscopy to investigate persistence to isoniazid in mycobacteria and observed that the phenotype represented a dynamic equilibrium between cell growth and stochastic expression of the pro-drug activating enzyme, katG. In addition, some antibiotics, such as FQs, can kill bacteria regardless of their growth status[14–16]. FQs corrupt DNA topoisomerases, including DNA gyrases, which have a vital role in replication and transcription[17]. As non-growing populations can be transcriptionally active[18,19], it is no wonder that FQs retain bactericidal activity against non-growing cells. Interestingly, a recent study using FACS found that ofloxacin (OFL) persisters within stationary-phase populations induce similar DNA damage responses as non-persisters, and that DNA repair machinery were not needed by persisters until after the treatment had concluded[16]. These data raised the possibility that persisters to FQs in growth-inhibited populations might be mutagenized from treatment, which could have significant implications for the development of genetically inherited resistance.

A paradigm in persistence research has been that survival is conveyed by limited corruption of the antibiotic primary target[4,9,13,20–22]. This paradigm is supported by studies on different persister types including those tolerant to β-lactams (tolerance by PBP inactivity or active efflux)[21], aminoglycosides (tolerance by limited influx)[22], and isoniazid (tolerance by lack of antibiotic activation)[13]. Importantly, a corollary of this common paradigm is that persistence would contribute to resistance development only through repeated rounds of population expansion. In other words, as persisters are essentially unscathed from antibiotic treatment, their offspring should only accumulate mutations through the processes of normal replication, which is not error-proof[23], and should not be more prone to resistance development than the offspring of untreated cells. Consequently, with multiple rounds of population reduction (antibiotic treatment) and rebound (survivor outgrowth), the cumulative likelihood of a resistant mutation to occur increases. Recently, the connection between persistence and resistance to β-lactams has been examined with cyclic adaptive laboratory evolution (ALE) experiments. Fridman and colleagues performed ALE experiments where E. coli were exposed periodically to ampicillin, and found that after 8 to 10 rounds, mutations that produced higher persister levels appeared[24]. Subsequently, it was shown that high-persister mutants provided a stepping stone toward resistance development through their ability to increase the number of bacteria (and thus genetic diversity) that survived antibiotic exposure[25]. Importantly, mutations that arose during the cyclic ALE experiments were attributed to errors associated with normal growth, and numerous rounds of β-lactam exposure were needed to observe the emergence of β-lactam resistance. This type of connection between persistence and resistance could have important clinical implications; however, additional links between these phenomena may exist. As stress responses implicated in resistance development are also important to persistence, some have speculated that the path from persistence to resistance may be more direct and faster than previously anticipated[26,27].

Volzing and Brynildsen previously reported that OFL persisters and non-persisters in stationary-phase cultures exhibited indistinguishable DNA damage responses, and DNA repair machinery and the SOS response were not needed until after the conclusion of treatment (recovery phase)[16]. Given these results, we hypothesized that a closer inspection of the recovery phase would be informative. Here using time-lapse fluorescence microscopy and an SOS reporter, we observe a total of 17 persisters and find that they generally induce impressive SOS responses and filament extensively before septating at multiple locations along their longitudinal axes. Many of the resulting daughter cells continue to replicate and resume normal appearances. Given these indicators of DNA damage, we test whether antibiotic-resistant mutants arise at a higher frequency from populations derived from OFL persisters compared to untreated controls. We find that mutants resistant to OFL, rifampicin (RIF), carbenicillin (CARB), D-cycloserine (DCS), and fosfomycin (FOS) are significantly enriched in OFL persister-derived populations. Focusing on RIF, we confirm that resistance is heritable, and that the enhancement observed is dependent on RecA, a cleavable LexA, an opportunity for persisters to recover from FQ treatment, and involves error-prone DNA polymerase V (UmuDC). These data show that after a single FQ treatment, the SOS response accelerates the development of resistance from FQ persisters to unrelated antibiotics. Such a direct connection between persistence and resistance differs from that delineated for β-lactams[25], and reinforces the importance of considering the antibiotic administered when studying persistence phenomena.

## Results

**OFL persisters filament and induce the SOS response**. We performed time-lapse fluorescence microscopy on wild-type MG1655 E. coli bearing a $P_{recA}$-gfp reporter plasmid as the population recovered from OFL treatment (Fig. 1). RecA induces the SOS response following DNA damage, is essential for homologous recombination, and the $P_{recA}$-gfp reporter was previously used to report on the SOS response and DNA damage[16,28]. As in the Volzing and Brynildsen study, we examined stationary-phase populations because stationary phase is the growth phase with the highest abundance of persisters, and growth-inhibited populations are the most difficult to treat with antibiotics[14]. The OFL treatment duration of 5 h (Supplementary Fig. 1a) and concentration of 5 μg/mL (Supplementary Fig. 1b and 1c) were chosen so that persisters would be the only

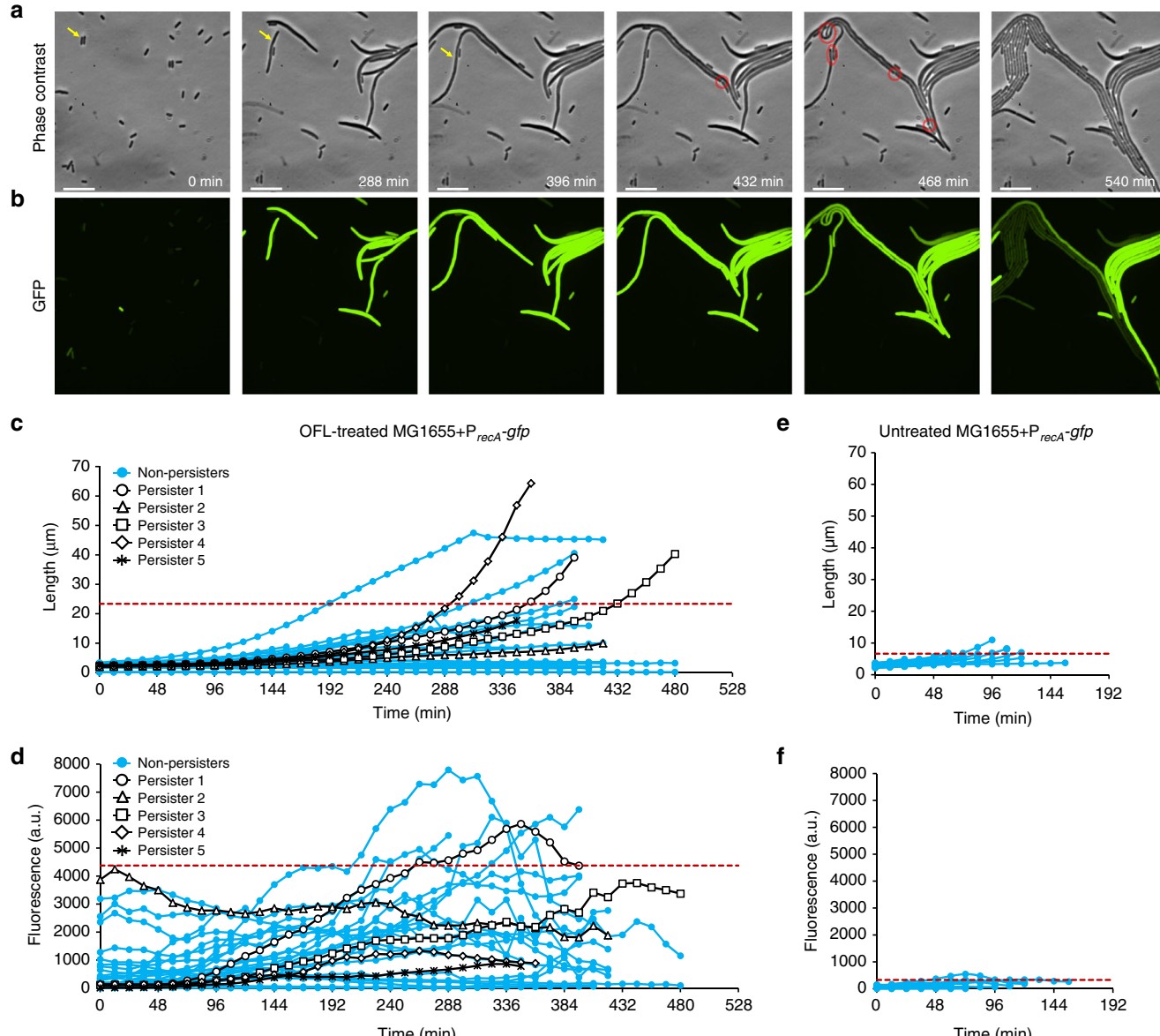

**Fig. 1** Time-lapse microscopy with *E. coli* MG1655. **a** Representative phase contrast and **b** GFP fluorescence images of recovering OFL-treated wild-type MG1655 *E. coli* bearing P$_{recA}$-*gfp*. Yellow arrow indicates persister. Red circles indicate division events. The far-right panel shows the incipient colony formation from a persister. Images are representative of six biological replicates. **c** Growth and **d** fluorescence of OFL-treated ($n = 30$) and **e**, **f** untreated cells ($n = 20$) recovering from treatment were analyzed. Of the 30 treated cells, five were persisters (black data points) and 25 were non-persisters (blue markers). 95% of the cell length and fluorescence measurements lie below the respective red dashed lines in **c**–**f**. The scale bars represent 10 μm. Refer to Supplementary Movies 1–3 for time-lapse videos

remaining CFUs, as we reported previously[16]. We stained nucleoids of the OFL-treated population with DAPI and observed that most cells were filamentous with diffused nucleoids located at mid-cell, which is consistent with previous reports of abnormal nucleoid morphology in DNA-damaged bacteria[29–31]. In addition, their morphologies resembled the physiology of bacteria treated with the DNA-damaging agent mitomycin C and were distinct from cells that filamented in response to a β-lactam, which provided evidence that the OFL treatment damaged DNA (Supplementary Fig. 2).

We monitored the recovery of OFL-treated MG1655 bearing P$_{recA}$-*gfp* (Fig. 1a–d, Supplementary Movies 1 and 2) along with an untreated control (Fig. 1e, f, Supplementary Fig. 3a, b, Supplemental Movie 3) and a promoterless reporter plasmid control (Supplementary Fig. 3c, d). In addition, we performed

controls where trace amounts of OFL (~0.2 ng/mL, ~30,000-fold reduction in concentration) were present during recovery, which were motivated by the incomplete removal of OFL from the washing procedure (please refer to Supplementary Methods for calculations). The presence of trace OFL did not induce filamentation or fluorescence (Supplementary Movie 4, Supplementary Fig. 3e, f). In all, we captured images of 17 persisters during their recovery from treatment. We used MicrobeJ[32] to quantify the growth and fluorescence of five of the persisters, which remained within the field of view throughout the duration of the time-lapse experiments. Phase contrast and fluorescence images depicting each of the other 12 persisters captured near the time of the first observed division event are shown in Supplementary Fig. 4. Figure 1a and b depicts one of the persisters that was tracked throughout the time-lapse experiment

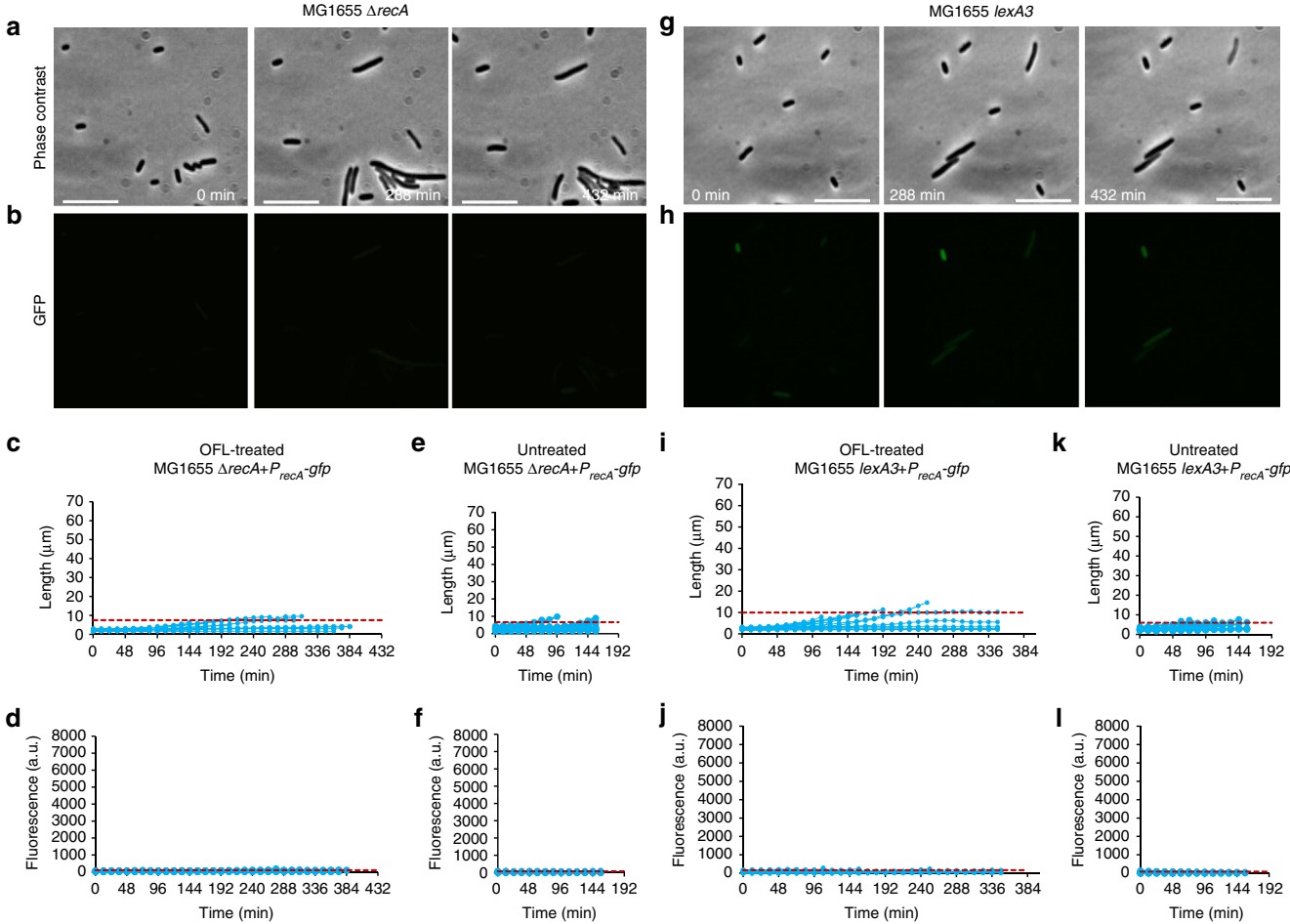

**Fig. 2** Time-lapse microscopy of *E. coli* MG1655 Δ*recA* and MG1655 *lexA3*. **a** Representative phase contrast and **b** GFP fluorescence images of recovering OFL-treated MG1655 Δ*recA* harboring $P_{recA}$-*gfp*. Growth and fluorescence of OFL-treated (**c**, **d**) and untreated (**e**, **f**) MG1655 Δ*recA* recovering from treatment were analyzed. **g** Phase contrast and **h** GFP fluorescence images of OFL-treated MG1655 *lexA3* harboring $P_{recA}$-*gfp*. Growth and fluorescence of OFL-treated (**i**, **j**) and untreated (**k**, **l**) MG1655 *lexA3* recovering from treatment were analyzed. 95% of the cell length and fluorescence measurements lie below the respective red dashed lines. Appreciable filamentation and fluorescence enhancement, and persister formation were not observed in either mutant. Fifteen cells from each treatment condition were quantified. The scale bars represent 10 μm. Refer to Supplementary Movies 5–8 for time-lapse videos

and is representative of the majority of the observed persisters, which filamented extensively and highly induced the SOS response. Untreated controls replicated normally and did not fluoresce appreciably, whereas the OFL-treated promoterless control filamented, but did not fluoresce, which confirmed that the fluorescence was dependent on the *recA* promoter.

We quantified the growth and fluorescence of 30 cells from the OFL-treated samples (Fig. 1c, d); these included five persisters, and 25 non-persisters that did not form microcolonies (representative of >100 cells imaged in six independent experiments). In addition, we analyzed 20 untreated cells all of which expressed very low levels of GFP and most of which divided around 60 min (Fig. 1e, f). Treated cells had a variety of outcomes. Of the 25 non-persisters analyzed, all eventually stopped growing and, for most, their fluorescence drastically decreased at the time growth ceased. At this time, the cells appeared to lose cytoplasmic contents and die. Although we found no consistent distinguishing feature between persisters and non-persisters based on level of expression from $P_{recA}$ or growth/elongation characteristics, the extensive filamentation (up to and beyond 60 μm) and SOS response induction in recovering OFL persisters are supportive of DNA damage. To assess whether these phenotypes depended on RecA and SOS induction, we performed analogous assays and analyses

with Δ*recA* and *lexA3* mutants carrying the reporter plasmid (Fig. 2, Supplementary Fig. 5, Supplementary Movies 5–8). The *lexA3* mutation encodes an uncleavable LexA[33], which prevents SOS induction. Notably, persisters were not observed with microcopy for either of these mutants, and treated cells failed to fluoresce and elongate to the same extent as wild-type cells. Specifically, 26.4% and 19.6% of the OFL-treated wild-type cell length data were above the 95% interval for the OFL-treated Δ*recA* and *lexA3* data, respectively (Fig. 2c, i); whereas, 88.6% and 82.7% of the OFL-treated wild-type fluorescence data were above the 95% interval for the OFL-treated Δ*recA* and *lexA3* data, respectively (Fig. 2d, j). Untreated Δ*recA* and *lexA3* mutants also failed to elongate and fluoresce (Fig. 2e–f, k–l). Collectively, these data indicate that OFL persisters experience DNA damage and use DNA repair mechanisms during recovery from treatment that enable their survival. These findings are consistent with and build upon previous work that used genetic constructs, flow cytometry, and FACS[16].

**High frequencies of resistant mutants from OFL persisters**. Considering that OFL persisters experience DNA damage, we reasoned that in some instances this damage or a cell's response

to it might lead to mutations. Further, these mutations might result in antibiotic resistance, which would establish a direct link between OFL persistence and resistance development. To test this hypothesis, we treated stationary-phase wild-type populations with OFL so that only persisters remained culturable and recovered those cells in fresh LB for 16 h, thus allowing them to repair the OFL-induced DNA damage and resume replication. Following recovery, approximately $10^9$ cells were plated on LB agar containing OFL (175 ng/mL), RIF (500 μg/mL), CARB (30 μg/mL), DCS (100 μg/mL), or FOS (20 μg/mL), and colonies were enumerated after 24 h of incubation at 37 °C. As Fig. 3a shows, the fold-changes between the number of resistant colonies to OFL, RIF, CARB, DCS, and FOS that arose from the OFL persister-derived population and those that arose from the untreated controls were significantly greater than a value of 1 (fold-change of 1 indicates equivalent abundances of resistant mutants). Raw values (number of resistant colonies per $10^9$ culturable cells) obtained from individual replicates are provided in Supplementary Fig. 6a.

We reasoned that the increase in antibiotic-resistant mutants in the persister-derived cultures could have resulted from enhanced survival of pre-existing resistant mutants during the OFL persistence assay, differences in genome amplification between the untreated- and persister-derived populations, or mutagenesis of OFL persisters during the recovery period that followed OFL treatment. First, we examined whether pre-existing OFL, RIF, CARB, DCS, or FOS mutants had higher survival to OFL in persistence assays. We harvested colonies resistant to 175 ng/mL OFL, 500 μg/mL RIF, 30 μg/mL CARB, 100 μg/mL DCS, or 20 μg/mL FOS that were derived directly from stationary-phase populations of wild-type *E. coli*, cultured those resistant colonies again to stationary phase, and subjected them to OFL persistence assays (Fig. 3b). The survivals of RIF-, CARB-, and FOS-resistant populations were not significantly different from the survival of the wild-type population, which provided evidence that the enhancement in resistance to these antibiotics was not associated with increased tolerance of their pre-existing mutants to OFL. On the other hand, the survivals of the OFL- and DCS-resistant mutants were significantly higher than the survival of wild-type by ~19.5- and 4.5-fold, respectively. Assuming equivalent growth in LB and negligible reversion of mutants to normal susceptibility, if the increases in resistant mutants following OFL treatment were due to the increased tolerance of pre-existing resistant mutants, we would expect to see at most a 19.5-fold increase in OFL-resistant mutants and a 4.5-fold increase in DCS-resistant mutants compared to untreated cultures (please see Supplementary Methods for calculations). With OFL, we observed a ~330-fold increase in resistance (Fig. 3a), which was not significantly different from 19.5-fold due to high variance in that data. However, for DCS we observed a 20.2-fold increase in resistant mutants, which was significantly higher than the 4.5-fold increase that would have been expected if only enrichment of pre-existing, DCS-resistant mutants was the sole cause of enhanced resistance. In consideration of these data and the widespread use of RIF resistance in the analysis of bacterial mutation rates[34,35], we focused the remainder of this study on RIF-resistance development.

Next, we evaluated the impact of trace amounts of OFL present in treated populations after washing and inoculation during the recovery period on RIF-resistance development. We inoculated the untreated control in LB with 0.02 ng/mL of OFL, where cells were grown for 16 h. The low dose of OFL present in the recovery media did not lead to increased RIF resistance (Supplementary Fig. 6b). We further considered whether the difference in culturable cells introduced into the recovery phase, which was ~50-fold greater for the untreated culture compared

to the OFL-treated sample, could explain the enhancement in resistance from OFL persisters. The reduced inoculum of culturable cells could enable the OFL-treated populations to undergo additional rounds of replication compared with the untreated population, thereby providing more opportunities to acquire mutations[35]. To assess whether this variable had any effect, we repeated the experiment and decreased the inoculum of untreated cells by 50-fold. We monitored the $OD_{600}$ and number of culturable cells at various time points during recovery (Supplementary Fig. 6c and 6d). We observed that while the $OD_{600}$ of the OFL-treated population increased continuously, CFUs in the OFL-treated population decreased 2 h following inoculation before increasing at later times. Therefore, we measured the number of RIF-resistant colonies that arose after 16 h in LB for both 50- and 100-fold reduced inoculums of untreated cultures, which approximate the initial and lowest CFU/mL of OFL persister-derived populations (Supplementary Fig. 6e). We observed that decreasing the inoculum size by 50-fold did not significantly increase the number of resistant colonies originating from untreated populations; however, 100-fold reductions produced ~2-fold more RIF-resistant mutants (Supplementary Fig. 6e). Importantly, when the levels of RIF-resistant mutants from OFL persister-derived populations were compared to 50- and 100-fold reduced inoculum untreated cultures, the enhancements in resistance development remained statistically significant and ~10-fold higher (Supplementary Fig. 6e).

To complement the reduced inoculum data, we measured total DNA at 0 and 16 h in untreated- and OFL persister-derived populations to quantify the extent of DNA amplification. We found that total DNA in both cultures at 0 and 16 h were comparable (Supplementary Fig. 6f), suggesting that the enhanced antibiotic resistance observed was not due to additional opportunities to synthesize DNA and acquire mutations. In addition, we observed that the DNA present in cell-free extracts collected at 0 h remained stable after 16 h of incubation at 37 °C (Supplementary Fig. 6g), suggesting that DNA released from lysed cells remained stable throughout the course of the experiment. We note that as LB contains DNA from yeast extract, which can interfere with DNA quantification from cells, we recovered untreated- and OFL persister-derived cells in EZ-rich media without glucose (MOPS-EZ media). Similar to LB, MOPS-EZ is amino acid-rich and lacks glucose. In populations recovered in MOPS-EZ, we observed increased resistance toward RIF in OFL persister-derived populations compared with untreated cells that was a similar magnitude to that in LB (Supplementary Fig. 6h), which suggested that RIF resistance was enhanced by OFL persisters in this media as well.

**Enhanced resistance development requires recovery from OFL.** In consideration of previous work where persisters to OFL did not need DNA repair until after the antibiotic was removed[16] and the time-lapse microscopy data of the recovery period illustrated here (Fig. 1), we suspected that OFL-induced mutagenesis of persisters occurred during the post-antibiotic recovery period. To test this hypothesis, we measured the abundances of RIF-resistant mutants in populations prior to OFL treatment and immediately following OFL treatment (no recovery period). As expected, the fraction of RIF-resistant mutants in those samples was indistinguishable (Fig. 3c), demonstrating that enhanced resistance development from OFL persisters could only be observed if they were allowed to recover from treatment. In addition, the data provided further confirmation that pre-existing RIF-resistant mutants did not have enhanced survival during the OFL treatment compared with wild-type cells.

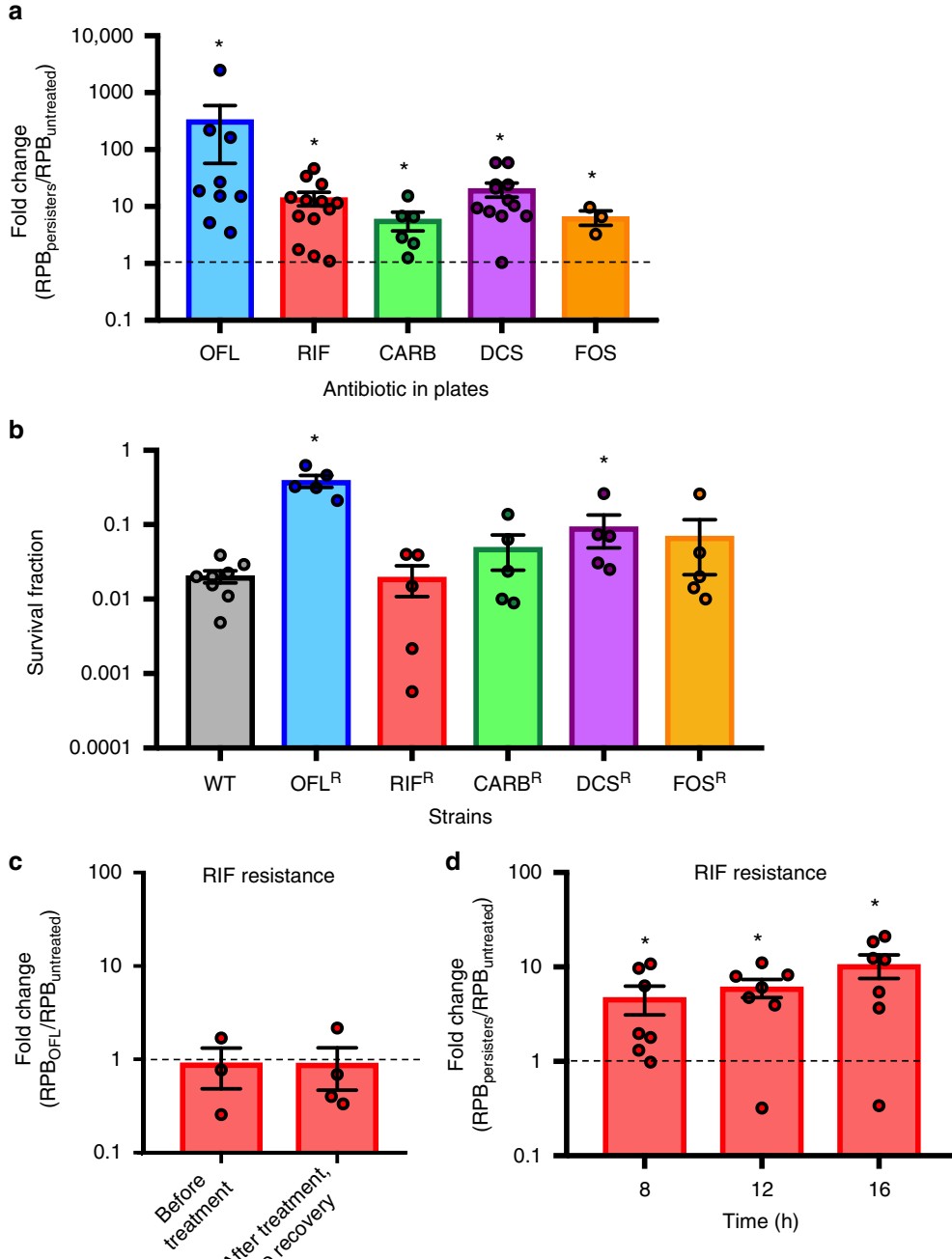

**Fig. 3** High frequencies of resistant mutants in cultures from OFL persisters. **a** The fold-changes between the number of resistant cells per $10^9$ culturable CFUs (RPB: resistant colonies per billion) to 175 ng/mL OFL ($n = 9$), 500 µg/mL RIF ($n = 13$), 30 µg/mL CARB ($n = 6$), 100 µg/mL DCS ($n = 12$), or 20 µg/mL FOS ($n = 3$) that arise from OFL persisters (RPB$_{persisters}$) compared to those that arise from untreated cells (RPB$_{untreated}$). Raw values for each independent replicate are provided in Supplementary Fig. 6a. **b** Wild-type ($n = 8$) and pre-existing populations of mutants resistant to 175 ng/mL OFL, 500 µg/mL RIF, 30 µg/mL CARB, 100 µg/mL DCS, or 20 µg/mL FOS (all $n = 5$) were cultured to stationary phase in LB and subjected to OFL persistence assays. RIF, CARB, and FOS-resistant mutants did not have a higher tolerance to 5 µg/mL OFL compared to wild-type, whereas OFL- and DCS-resistant mutants exhibited higher tolerances. **c** Without recovery in an antibiotic-free environment, enhancement of RIF-resistant mutants from OFL persisters is not observed. Fold-change represents RIF-resistant mutants derived from OFL-treated populations (RPB$_{OFL}$) and untreated populations (RPB$_{untreated}$) before antibiotic addition ("Before Treatment", $n = 3$) and after OFL removal at the end of 5 h of treatment ("After treatment, no recovery", $n = 4$). **d** Time course analysis of RIF-resistance development during recovery reveals that an enhancement in resistance can be observed as early as 8 h of recovery from OFL treatment ($n = 7$). We note that $10^9$ culturable CFUs could not be obtained at 2 and 4 h of recovery in LB, and thus resistance could not be quantified at those earlier time points. Error bars portray S.E.M. For **a**, **c**, and **d**, a value of 1 is indicative of equivalent abundances (no change). Asterisks in these panels denote a log-transformed fold-change that is significantly different compared to a log-transformed value of 1 ($p \leq 0.05$, using two-tailed $t$-tests with unequal variances). For **b**, asterisks denote statistically significant difference in survival fraction compared with the wild-type

To discern the time during recovery at which populations from OFL persisters exhibited increased abundance of resistant mutants, we performed experiments at 8 and 12 h of growth in LB. Significant enhancements were observed from 8 h and onward (Fig. 3d). We note that we attempted to perform analogous experiments at 2 and 4 h in LB, but did not have sufficient cells in OFL persister-derived cultures to reach $10^9$ cells for plating on RIF-containing plates. These data indicate that the enhancement in RIF resistance from OFL persisters arises at or before 8 h of recovery in LB media.

**Role of RecA, SOS response, and UmuDC in enhanced resistance.** Given the importance of the post-OFL recovery period to the enhancement of resistance from OFL persisters, we hypothesized that RecA would be required to observe that phenomenon. To test this hypothesis, we assessed RIF-resistance in Δ*recA* OFL persisters and untreated controls. As depicted in Fig. 4a, without *recA*, an enhancement in RIF resistance was no longer observed. Complementation of Δ*recA* restored the enhancement of RIF-resistant mutants that arose from persister-derived populations (Fig. 4a; Supplementary Fig. 7a). As RecA participates in recombinatorial repair and SOS induction, we tested a *lexA3* mutant to more specifically probe the role of SOS induction. As depicted in Fig. 4b, *lexA3* mutants do not exhibit increased levels of RIF-resistant mutants in populations derived from their OFL persisters (Supplementary Fig. 7a contains raw data). Collectively, these data confirm that *recA* and SOS induction are essential for the enhanced development of RIF resistance from OFL persisters. We also note that the fold-changes in RIF

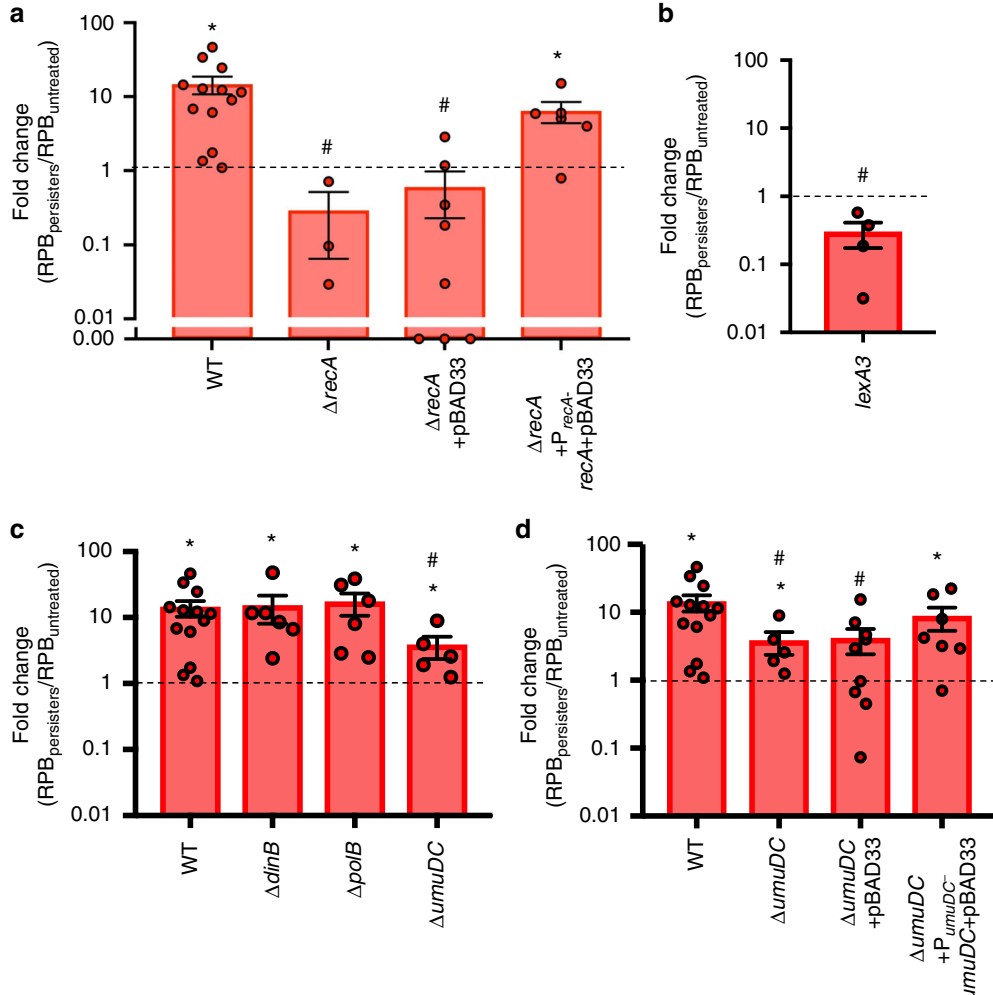

**Fig. 4** Resistance enhancement involves RecA, SOS induction, and UmuDC. **a** OFL persisters derived from Δ*recA* ($n = 3$), which is defective in SOS induction and recombinatorial repair, do not give rise to a population with increased RIF resistance. Complementation of Δ*recA* restores the enhancement of RIF-resistant mutants from OFL persisters ($n = 6$), whereas persisters from Δ*recA* mutants bearing the pBAD33 control do not give rise to a population with enhanced RIF resistance ($n = 8$). **b** Increase in RIF-resistant mutants was not observed in OFL persisters derived from *lexA3* mutants ($n = 4$), which cannot induce SOS. **c** Upon deleting *umuDC*, which encodes error-prone, SOS-induced DNA polymerase V, a significant increase in RIF-resistant mutants in OFL persister-derived populations was observed ($n = 5$), but the magnitude of that increase was significantly lower than that of wild-type ($n = 13$). Mutants lacking error-prone polymerase II (Δ*polB*, $n = 6$) or error-prone polymerase IV (Δ*dinB*, $n = 6$) had significant increases in RIF-resistant mutants in OFL persister-derived populations that were of a comparable magnitude as wild-type. **d** Complementation of UmuDC on pBAD33 in Δ*umuDC* mutants restores increase in RIF-resistant colonies from OFL persisters ($n = 7$), whereas the presence of pBAD33 control fails to restore the phenotype ($n = 9$). Error bars portray S.E.M. * indicates significance ($p \leq 0.05$) between log-transformed fold-changes and log-transformed value of 1 (fold-change of 1 would indicate no difference in resistant mutants). # indicates significance ($p \leq 0.05$, using two-tailed *t*-tests with unequal variances) between log-transformed fold-changes and those of wild-type. Zero values for Δ*recA*+pBAD33 were not included in the statistical analysis because they could not be log-transformed

resistance for $\Delta recA$ and $lexA3$ were lower than 1 ($p = 0.16$ and $p = 0.08$, respectively), which if significant would have reflected reduced resistance development and an interesting avenue for clinical extrapolation of these findings.

The SOS regulon consists of genes that encode three SOS-inducible polymerases, Pol II ($polB$)[36], Pol IV ($dinB$)[37], and Pol V ($umuDC$)[38], which participate in DNA synthesis in response to DNA damage[39] and long-term stationary-phase adaptation[40,41]. We measured RIF-resistant colonies arising from mutants harboring deletions of each of these polymerases (Fig. 4c; Supplementary Fig. 7b) and observed that the deletion of each polymerase resulted in statistically significant increases in RIF-resistance development between populations arising from OFL-derived persisters compared with untreated controls. Enhancements for $\Delta polB$ and $\Delta dinB$ did not differ significantly from that of wild type (Fig. 4c), whereas a significantly reduced enhancement in RIF resistance compared to wild type was observed for $\Delta umuDC$. Complementation of $umuDC$ restored the enhancement to wild-type levels (Fig. 4d; Supplementary Fig. 7c). These results implicate the SOS-inducible and error-prone Pol V in increased RIF resistance from OFL persister-derived populations. However, we note that the significant enhancement observed in $\Delta umuDC$, though lower in magnitude than that of wild-type, suggests that additional processes are involved.

**OFL persisters generally repair DNA damage correctly**. Although the incidences of RIF-resistant mutants were ~14-fold higher in populations spawned from persisters compared to controls, the absolute frequency remained low (~100 or more in $10^9$ cells). Interestingly, all 17 OFL persisters that we observed experienced DNA damage, as inferred from filamentation and SOS induction (Fig. 1, Supplementary Fig. 4, Supplementary Movies 1 and 2), and filaments were observed to septate numerous times and produced multiple viable progeny. Given this physiology, we reasoned that mutations in general, not just those conferring resistance to antibiotics, might be prevalent in the progeny of OFL persisters. To assess this possibility, we isolated DNA from single colonies grown from OFL persisters or untreated controls, and performed whole-genome sequencing (WGS). We elected to portray the WGS results using nucleotides with reference allele frequencies (RAF) less than or equal to indicated values instead of setting a threshold for calling mutations, because there was potential for genetic heterogeneity in persister colonies (Fig. 1a). When WGS data from 10 persisters, 10 untreated controls, and 4 parental cultures were analyzed, RAFs at different thresholds were indistinguishable (Fig. 5a). Additional analyses of the WGS data are presented in Supplementary Fig. 8a–c, and from these we do not observe any discernable genomic differences between persister and untreated control colonies whose genomes were sequenced. These data suggest that persisters, despite experiencing DNA damage, generally repair the damage correctly. Importantly, given the frequency of RIF mutants in persister-derived populations, which is ~$1 \times 10^{-7}$, it is not surprising that we did not observe mutations that confer RIF resistance in the sequencing results of 10 colonies derived from persisters.

To confirm that the RIF resistance observed in this work was due to heritable genetic changes, we performed WGS on RIF-resistant colonies derived from parental cultures, untreated controls, and OFL persisters (Supplementary Table 1). We observed that all colonies derived from parental cultures and persisters, as well as six of the eight colonies from the untreated controls, harbored mutations that mapped to $rpoB$, the common locus for RIF-resistant mutations.

**Most stationary-phase cells have two or more chromosomes**. Homologous recombination is the primary method that $E.\ coli$ uses to repair DNA double strand breaks[42]. Given the OFL concentration used here (80-fold higher than the MIC) (Supplementary Fig. 1), DNA double strand breaks will be the prevalent form of damage[17]. To repair such DNA damage with high fidelity, persisters would need more than one copy of the broken loci to perform homologous recombination. As the cultures examined here were treated well into stationary phase, and thus were not undergoing DNA replication, we sought to quantify the chromosome distribution of those cultures. Using PicoGreen, which is a highly specific double-stranded DNA dye[43], we found that ~69% of the population contained two or more chromosomes (Fig. 5b and Supplementary Fig. 8h; Supplementary Fig. 8i for chromosome number controls[44]). Therefore, the majority of stationary-phase cells had the genomic content necessary to perform homologous recombination, even though only a fraction of them survived the DNA damage imposed by OFL treatment. Furthermore, loss of $ruvA$ or $recG$, which resolve recombinational intermediates during homologous recombination[45], exhibited ~16-fold and ~29-fold lower persister levels than wild-type, respectively, and $\Delta recG\ \Delta ruvA$ exhibited highly reduced persister levels that approached those of $\Delta recB$ (Fig. 5c) and $\Delta recA$[16].

**Ciprofloxacin persisters accelerate resistance development**. We sought to assess whether the enhancement in antibiotic resistance development from persisters was specific to OFL, or shared with additional FQs. We performed identical experiments with 1 μg/mL of CIPRO instead of OFL, and resistance to RIF was assayed. As depicted in Fig. 5d and Supplementary Fig. 7d, the acceleration of resistance development in the progeny of persisters compared to controls occurred with CIPRO as well, which establishes that the phenomenon is shared between distinct FQs.

## Discussion

Antibiotic failure in any form is cause for concern, regardless of whether it results from resistance, tolerance, or persistence. Persistence is particularly disheartening because it causes failure under best-case treatment scenarios: the bacteria are genetically sensitive to the drug and within conditions where most bacteria are killed. An added complication is that not all persisters are created equal[1,3,46], with notable examples including the discovery of TisB as a genetic determinant of CIPRO persistence (but not β-lactam or aminoglycoside persistence) in exponential phase $E.\ coli$ cultures[7,8], and the identification of shared and distinct mediators of ampicillin and OFL persister formation in response to carbon source transitions[5,6,47]. The data presented here provide direct evidence that persisters are heterogeneous, and that consideration should be given to the mechanism of action of the antibiotic when studying this phenotype. β-lactam persisters have been imaged with microscopy previously[9], and they exhibited little response to the antibiotic during treatment and what appeared to be normal growth resumption following treatment. In contrast, the FQ persisters imaged here exhibited a response following treatment that was indicative of considerable DNA damage (Fig. 1). Interestingly, a recent investigation of a model persistence system based on accumulation of the MazF toxin depicted filamentation following the conclusion of CIPRO treatment[48]. Data presented here show that similar morphological changes occur in wild-type $E.\ coli$ persisters following OFL treatment.

With direct evidence of DNA damage and SOS induction in OFL persisters, we performed WGS of persisters and untreated controls and assayed for antibiotic resistance. WGS indicated similar levels of mutagenesis between OFL persisters and untreated controls when 10 colonies of each were sequenced

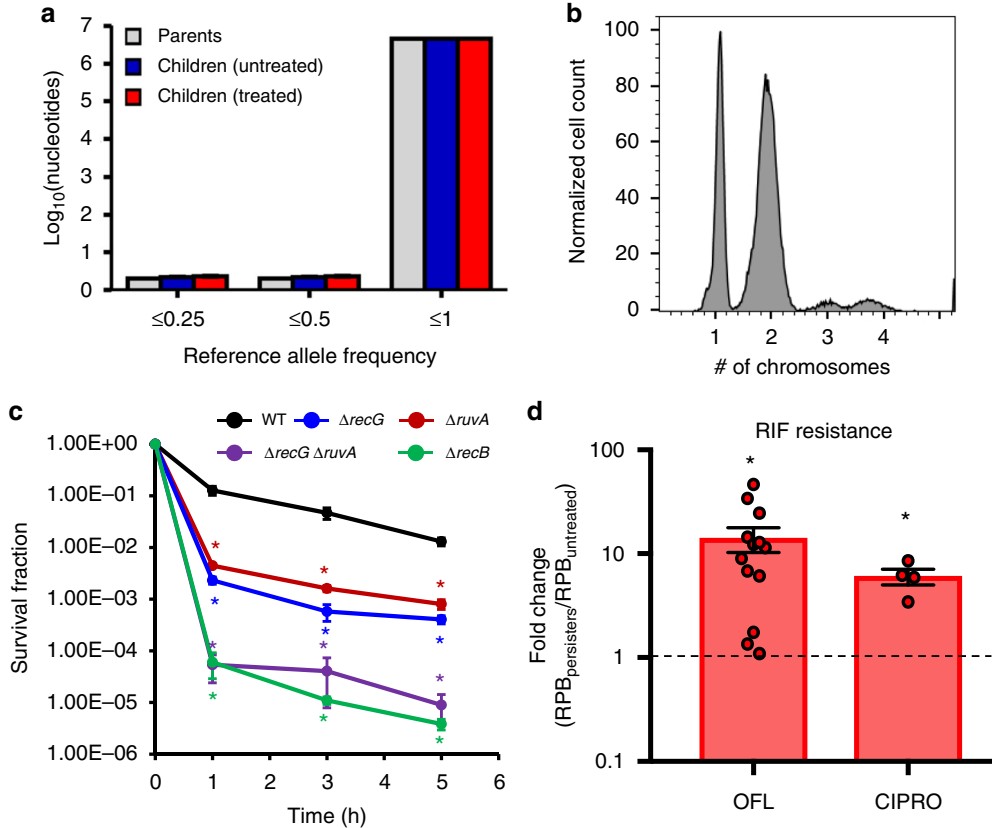

**Fig. 5** Fidelity of DNA repair in OFL persisters. **a** WGS was performed from DNA extracted from 10 persisters, 10 untreated controls, and 4 parental cultures, and number of nucleotides with reference allele frequencies (RAF) less than or equal to the indicated values are depicted in the histogram. RAF of 1 indicates that all sequencing reads of that nucleotide were of the reference nucleotide, whereas RAF of 0 indicates that all sequence reads had a non-reference nucleotide. Values between 0 and 1 indicate that some reads were of the reference nucleotide. The RAFs at different thresholds were indistinguishable among the persisters, untreated controls, and parental cultures, suggesting that OFL persisters generally repair DNA damage correctly. **b** Chromosomal content of stationary-phase cultures prior to OFL treatment was determined using PicoGreen stains ($n = 3$; see Supplementary Fig. 8i for controls used to determine fluorescence intensities corresponding to chromosome number). $30.8 \pm 1.6\%$ of cells contained one chromosome, whereas $68.4 \pm 1.6\%$ of cells contained two or more chromosomes, indicating that the majority of cells had the chromosomal content for repair by homologous recombination. Histogram shown is representative of three biological replicates. **c** Mutants defective in homologous recombination, which requires the resolution of Holliday junctions by RuvA and/or RecG, demonstrate reduced survival to OFL treatment in stationary phase ($n = 3$). **d** Enhancement in RIF resistance was observed in progeny of CIPRO persisters ($n = 4$), similar to offspring of OFL persisters, demonstrating that the phenomenon of resistance enhancement following treatment and recovery is shared between distinct fluoroquinolones. Error bars depict S.E.M. For **c**, * indicates significance between log-transformed survival fractions and those of wild-type ($p \le 0.05$, using two-tailed $t$-tests with unequal variances). For **d**, a value of 1 is indicative of equivalent abundances (no change), and * denotes a log-transformed fold-change that is significantly different compared to a log-transformed value of 1 ($p \le 0.05$, using two-tailed $t$-tests with unequal variance)

(Fig. 5a). These data directly confirmed that most OFL persisters conducted high-fidelity DNA repair, which was consistent with previous evidence and data presented here (Fig. 5c) of the importance of homologous recombination for their survival[16]. Resistance assays revealed enhanced abundance of resistant mutants in the progeny of OFL persisters, which included those that were resistant to different antibiotic classes (RIF, CARB, DCS, and FOS) (Fig. 3a). Further, we found that this enhancement was dependent on RecA, SOS induction, and a chance to recover from OFL treatment (Fig. 3c, d, Fig. 4a, b), and involved the error-prone polymerase UmuDC (Fig. 4d). Potential implications of these findings include that efficacious treatment of bacterial populations with FQs, which yield persisters as the only bacteria left that can regrow, can produce resistant mutants not only to FQs themselves but also to antibiotics with completely different mechanisms of action. If the population is not eradicated with the first round of treatment, the ability to treat it even with other antibiotics could decline.

Examples of how antibiotics accelerate resistance development beyond their role in selecting for it have been revealed in the past few years, and our work adds to this knowledge base by demonstrating a direct connection between persisters and resistance. Almost uniformly, resistance enhancement from antibiotic exposure has been associated with concentrations around or below the MIC[49]. For instance, Thi and colleagues[50] observed accelerated resistance development in *E. coli* treated with antibiotics around their MICs through the process of SOS-induced mutagenesis, which involves the action of LexA-controlled error-prone DNA polymerases[51,52]. Using single-cell techniques, Bos and coworkers observed that CIPRO at eightfold less than the MIC stimulated elongation of *E. coli* and budding from the cell poles[53]. Importantly, some of the buds were resistant to CIPRO and began to grow, whereas the filaments from which they originated died. Two additional studies examined the development of CIPRO resistance following prolonged exposure (over days) to CIPRO around twofold above the MIC on agar[54,55]. In those

studies, resistant colonies were removed from pads as they arose and new resistant colonies were observed after the time period associated with when pre-existing mutants would form colonies (2 days), which suggested mutagenesis of susceptible bacteria on the plates[54,55]. Compared with these studies, however, we used concentrations ~80-fold higher than the MIC, which are more damaging to DNA[17], and whose associated persister levels do not vary appreciably with concentration (Supplementary Fig. 1c). Under our experimental conditions, we also observed enhancement of resistance toward RIF, CARB, DCS, and FOS following recovery from treatment with supra-lethal doses of FQs. Results from these previous works and our present study suggest that FQs can accelerate resistance development over an exceptionally broad range of concentrations and to unrelated antibiotic classes.

Data presented here provide greater understanding of FQ persistence, yet also stimulate more questions. With comparable SOS induction and filamentation, it remains unclear as to how FQ persisters survive such damage whereas the majority of their kin succumb to it. Some possible hypotheses include the location of such damage on the genome and/or the number of genomes in individual cells. Recovery from treatment, SOS induction, and Pol V were involved in accelerated resistance development from FQ persisters, but it remains unclear how long after treatment concludes that resistant mutants appear because at the earliest time point we could measure RIF resistance had already significantly increased. It also remains to be fully delineated how similar or different antibiotic-resistant mutants from untreated and FQ-persister populations are and whether any differences impact the enhancement of resistance from FQ persisters. As these and other questions are answered by future works, a more detailed understanding of FQ persistence will emerge and novel strategies to eliminate these survivors envisioned. Overall, this investigation shows that FQ persisters are far from minimally-impacted bystanders of treatments, and that it is their struggle to survive FQ-induced damage that enables them to produce resistant mutants at accelerated rates.

## Methods

**Bacterial strains and plasmids**. Bacterial strains and plasmids in this work are listed in Supplementary Table 2. Primers used for strain construction are listed in Supplementary Table 3. Procedures for strain construction are detailed in the Supplementary Methods.

**Persistence assays**. Antibiotic tolerance assays were carried out with stationary-phase cultures that were grown for 20 h in M9 media with 10 mM glucose as the sole carbon source (with 50 µg/mL kanamycin or 25 µg/mL chloramphenicol for plasmid maintenance where indicated). Cultures were treated with 5 µg/mL OFL or 1 µg/mL CIPRO for 5 h. CFUs were enumerated from samples collected prior to antibiotic administration and periodically during treatment. See Supplementary Methods for further details.

**Resistance assays**. To quantify the development of antibiotic resistance from persisters, persistence assays were carried out with stationary-phase cultures as described above. After removing OFL or CIPRO, persisters were recovered in LB (with 25 µg/mL chloramphenicol for plasmid maintenance where indicated) or when indicated MOPS-EZ Rich media (Teknova EZ Rich defined media lacking glucose) for 16 h and ~10⁹ cells were plated on LB agar with 175 ng/mL OFL, 500 µg/mL RIF, 30 µg/mL CARB, 100 µg/mL DCS, or 20 µg/mL FOS to enumerate resistant colonies. Detailed descriptions are provided in the Supplementary Methods.

**Time-lapse microscopy**. Growth and fluorescence of OFL-treated cells and untreated controls during recovery from antibiotic treatment on LB agar supplemented with 50 µg/mL kanamycin (KAN) for plasmid retention, as well as untreated cells recovered on LB-KAN agar containing ~0.2 ng/mL OFL, were monitored using a fully motorized Nikon Ti-E inverted microscope with Perfect Focus System equipped with Yokogawa spinning disc (CSU-21). Sample preparation, microscope setup, and image analysis are detailed in the Supplementary Methods.

**Statistical analysis**. All replicates were independent biological replicates. Significance was assessed using two-tailed *t*-tests with unequal variances on log-transformed values and a *p*-value threshold of 0.05.

**Reporting summary**. Further information on experimental design is available in the Nature Research Reporting Summary linked to this article.

## Data availability
Sequencing data that support the findings of this study have been deposited in the BioProject Database with the accession code of PRJNA517575. Additional data that support the findings of this study are available from the corresponding author upon reasonable request.

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

## Acknowledgements

We thank Zemer Gitai, Ned Wingreen, Thomas Silhavy, Katherine Volzing, Maxwell Wilson, Benjamin Bratton, Wei Wang, Jessica Wiggins, Ling Guo, Jennifer Miller, Daniel Sanchez, Lance Parsons, Gary Laevsky (Nikon Center of Excellence in the Princeton Confocal Imaging Facility), Christina DeCoste (Princeton Flow Cytometry Resource Center), and Sandra Aedo for assistance and advice. We thank the National BioResource Project (NIG, Japan) for distribution of the Keio collection. This work was supported by the NIAID of the NIH (T.C.B.: F30AI114163, M.P.B: R21AI115075, R01AI130293), the Charles H. Revson Foundation (W.W.K.M.: Fellowship in Biomedical Science), and Princeton University (M.P.B.: startup funds). This content is solely the responsibility of the authors and does not necessarily represent the views of the funding agencies.

## Author contributions

T.C.B., W.W.K.M., A.M.M. and M.P.B conceptualized and planned experiments. T.C.B., W.W.K.M. and A.M.M. performed the experiments. T.C.B., W.W.K.M., A.M.M. and M.P.B. analyzed the data and wrote the paper.

## Additional information

**Competing interests:** The authors declare no competing interests.

