## [Peer Review File · Nature Communications]

Reviewers' comments:

Reviewer #1 (Remarks to the Author):

There are essentially four key points made in the manuscript by Barrett et al. 1) offspring emanating from bacterial persisters (treated with fluoroquinolone) gave rise to more antibiotic-resistant mutants than untreated bacteria; 2) resistance was heritable; 3) resistance depended generally on SOS induction and more specifically on DNA polymerase V; 4) it appears that cells resistant to antibiotics needn't proceed through a tolerance phase described by Balaban et al. (Ref. 8). I believe that ideas and data presented are fresh and important. I have several questions and comments that I'd like the authors to consider.

1. The requirement for pol V in generating antibiotic resistance measured as increased numbers of RIF-resistant colonies emanating from OFL-treated cells (red bars) relative to untreated cells (grey bars), is shown in Fig. 3F. While all three pol mutant strains show a drug-dependent increase in RIF colonies, the differences for Δ umuDC aren't statistically different. Based principally on these small statistical differences, the authors conclude that the increase in mutagenesis of treated cells is dependent specifically on the presence of pol V, and not pols II or IV. This conclusion is reinforced by the restoration of mutagenesis when pol V is overexpressed in a Δ umuDC background (Supp. Fig. 3). Even so, the data are less than overwhelming.

However, let's examine a different comparison. The wild type control data are shown in Fig. 3A, in which the first pair of bars are untreated (dark gray) and OFL treated (red). Additional data are shown in Fig. 3D and Fig. 3E for the *recA* and *lexA3* mutants, respectively. First, compare the data for the untreated cells (dark grey). The wild-type strain yields ~10 RIF colonies per 10^9 CFU. While all the mutant strains yield roughly 30 RIF mutant colonies per 10^9 CFU (Fig 3D and 3F).

Why do each of the untreated pol mutant strains give rise to about three times more mutants in the absence of treatment comparing to wild-type untreated strain?

Now let's examine the surviving persisters from the treated cells. The WT (Fig. 3A) yields about 100 RIF colonies per 10^9 CFU on average, which is similar to Δ polB, Δ dinB, and Δ umuDC deletion strains (all yield ~100 RIF colonies per 10^9 CFU). So, yes, if you compare effect for persisters of each mutant strain to untreated mutant control strains, Δ umuDC looks like it might have a reduced effect. However, if instead a comparison is made to treated wild type and treated mutants (which perhaps might be a more relevant way to view the experiment, i.e., same treatment but different strain), there is little if any difference in the number of RIF resistant mutants (compare red bars in Fig. 3A and 3F).

Could the authors comment on this possible alternative interpretation?

2. Regarding the Δ recA (Fig. 3D) and lexA3 (Fig. 3E) data. The recA deletion strain appears to be an antimutator following drug treatment. While there is no asterisk shown, might 8 (drug treated) vs 30 (untreated) colonies reflect an actual difference?

For the lexA3 strain, there is virtually no difference in mutants between treated and untreated cells, but again, compared to WT, both have significant reductions compared to wild type in treated cells.

3. With respect to the induction of the recA-GFP reporter and its relationship to cell length, as shown in Fig. 1B, I see no relationship between GFP induction and filamentation (the proxy for being a persister.) The five persisters are shown in black and, you can see their lengths over time (Fig. 1B, black). I've tried to look at their respective levels of GFP induction. It's not easy to do as presented, but if one looks hard, it's possible to determine which black line in the lower panel corresponds to which black line in the upper panel, because for each cell the data are acquired for different lengths of time. (I strongly suggest that that different shapes and/or colors be used for the five persisters, so that the correlation can be made much more easily.) The problem is that there does not appear to be a correlation in the amount of GFP expression and the time the cell is a persister. If you put the persisters in rank order in length from longest to shortest, 1, 2, ..., 5, and then compare them to the rank order of their GFP levels (i.e., amount of RecA induction), the longest ranks 4th, the next 1st, then 2nd, then 4th, then 3rd. However, an even more important issue would seem to be that many of the non-persisters (blue lines) appear to show GFP levels greater than the persisters.

4. A general comment on mutation is made in line 241 ("mutations in general, not just those conferring resistance to antibiotics, might be prevalent in the progeny of OFL persisters"). Therefore, the higher the level of general mutation, the better the chance for increased antibiotic resistance. The mutation frequency for RIF is $10e^{-7}$, which would seem to be in the range of typical mutation frequencies measured in batch culture. Presumably, the authors would attribute this seeming equivalence to the accurate repair of most DNA polymerase-catalyzed misincorporations. Alternatively, however, might it be reasonable to suggest that there is perhaps no transient increase in mutation frequency, but rather the level of mutations might instead be the baseline frequency expected for a phenotype having a large base pair target size?

5. In several places in the manuscript, the authors refer to "extensive" or "massive" amount of DNA damage the cells have experienced. However, the amount is not quantitated, and I don't think they actually do try to link it to GFP expression directly. However, the authors do imply that if cells are filamenting, then there must be a large amount of DNA damage. Is that actually true? Wouldn't a cell with a single DSB continue to filament until the damage is repaired? And, if one-third of the cells only have one chromosome, and the authors suggest that two chromosomes are needed for homologous recombination-based repair, then perhaps those cells having a single chromosome might filament for a long time. I suggest that the authors comment on this point.

6. Suggestions concerning the references:

a. For the experiment where the number of chromosomes per cell are determined (line 264), the work of Bernander and colleagues (Supplement Ref. 19) should be cited in the text references. In Supplement Ref. 19, Akerlund et al. showed that stationary phase *E. coli* can have 1, 2, 4 or 8 chromosomes. (Barrett et al., report 1, 2, 3, and 4.) The authors should explain where cells with 3 chromosomes come from. Generally, it is thought that if the origins are going to fire multiple times, they fire together. At least that appears to follow from Akerlund et al., since they did not observe 3 or 5 or 6 chromosomes, just 1, 2 and powers of 2.

b. Reference 31 for the encoding of pol II should be replaced by the primary citation: Bonner, C.A., Hays, S., McEntee, K., and Goodman, M.F. DNA Polymerase II is Encoded by the DNA Damage-Inducible *dinA* Gene of *Escherichia coli*. *Proc. Natl. Acad. Sci. USA* 87, 7663-7667 (1990).

c. Reference 33 for the encoding of pol V should be replaced by the primary citation: Tang, M., Shen, X., Frank, E. G., O'Donnell, M., Woodgate, R., and Goodman, M. F. UmuD'2C is an error-prone DNA polymerase, *Escherichia coli* pol V. *Proc. Natl. Acad. Sci. USA* 96, 8919-8924 (1999).

d. On line 222 the authors refer to the "three nonessential polymerases." But these three pols are non-essential in monoculture. They are essential for long-term survival in co-culture as shown in Yeiser, B., Pepper, E. D., Goodman, M. F., and Finkel, S. E. SOS-induced DNA polymerases enhance long-term survival and evolutionary fitness. *Proc. Natl. Acad. Sci. USA* 99, 8737-8741 (2002). As a closely related point: the authors say that the three pols "participate in DNA synthesis in response to DNA damage" (Line 223-224). That's correct, but the three pols also participate in DNA synthesis in the absence of exogenous DNA damaging agents as shown in Corzett, C. H., Goodman, M. F. and Finkel, S. E. Competitive Fitness during Feast and Famine: How SOS DNA Polymerases Influence Physiology and Evolution in *Escherichia coli* *Genetics* 194, 409-420 (2013).

Reviewer #2 (Remarks to the Author):

This manuscript by Barret et al examines how a single exposure of persister cells with fluoroquinolones affect the frequency of resistant mutants using time-lapse microscopy, mutation frequency measurements and genetic analysis. The authors show that FQ induces the SOS response and suggest that this results in a strong increase in mutation frequency in the recovered persister population.

The most important experiments in this manuscript are the measures of mutation frequencies. Unfortunately, they are not fully described and they also lack some key controls that I think are needed before one can conclude what the authors suggest is happening (i.e. 100-fold increase in mutation frequency in persisters in response to FQ exposure)

Major points

1. The mutation frequency assay is not as clearly described as it should be. Key information is lacking regarding how many cells were inoculated into what volume from the fluoroquinolone and untreated control? Thus, to make a fair comparison of mutation frequencies between treated and untreated populations one needs to know the number of generations of growth for each population (see also point 2 below).
2. Another important point regarding the mutation frequency measurement is to what extent during the recovery phase, the death rate is higher in the treated population as compared to the non-treated. Thus, if there is a higher death rate in the treated population (for example due to the accumulated DNA damage or presence of intracellular residual FQ) one will underestimate the number of generations of growth and the mutation frequency will appear higher (even though there is in fact no increase in the intrinsic mutation rate). A recent and very important paper from the Bonhoeffer laboratory shows that this generally results in overestimation of mutation rates during antibiotic exposure. Thus, I would strongly suggest that the authors measure the number of divisions with the plasmid system used in Frenoy and Bonhoeffer (PLoS Biol. 2018, 16(5):e2005056) to examine/exclude this possibility.
3. It is interesting to note that in other studies, LexA defective mutants with constitutive SOS induction show a very moderate (a few-fold) increase in mutation rates. How does that square with the present observation of a 100-fold increase? Does it not suggest that there are other factors than the SOS induction per se (and an associated increase in PolV) that cause the apparent increase in mutation frequency? It would be interesting to see if artificial over-expression of PolV (to the same level as that seen during SOS induction) results in the same level of mutagenesis.
4. One of the authors' main points is that persisters might contribute to evolution of acquired resistance during treatment. But to be fair in that analysis should one not take into account the relative contributions from the non-persisters and persisters with regard to mutant formation? That

is, the likelihood that resistant mutants will appear in the respective populations is: fraction of total cells x relative mutation rate. For normal cells this would: $0.9999 \times 1 = 0.9999$ and for persisters: 0.0001 (assuming that the persister fraction is $10E-4$) $\times 100 = 0.01$, indicating that even with a 100-fold increase in mutation rate in the persister population the large majority of mutants are still derived from the normal population. I think this ought to be explicitly discussed.

Other comments

5. Line 33. What is actually the evidence that they are not genetically different?
6. Line 34. "Favorable phenotypic niche? Sounds nice but what does it mean?"

Reviewer #3 (Remarks to the Author):

This is a very elegant study showing that fluoroquinolone persisters develop a strong SOS response leading to the formation of antibiotic-resistant mutants. This effect is mediated by RecA and mutagenic polymerases. The manuscript is straightforward and logically structured. This work is novel and may have important consequences for the development of antibiotic-resistance in a clinical context.

Please find below my detailed comments.

General remarks

- Despite the washing steps performed after OFL treatment, low OFL concentrations (i.e. $0.002 \mu\text{g/ml}$ or $1/32x$ MIC) remain present in the samples. A control experiment should be performed where untreated cells are put on an agar pad containing $0.002 \mu\text{g/ml}$ OFL, to confirm that this concentration does not induce the observed phenotypic effects of filamentation and SOS response activation. Similarly, an experiment could be performed where untreated cells are put in LB containing $0.002 \mu\text{g/ml}$ OFL for 16h and plated onto RIF plates, to exclude the potential effect of subinhibitory OFL concentrations on the mutation rate (as reported by Kohanski et al., 2010).

- The authors claim that the 16h recovery phase is required for the enhancement of RIF resistance. However, the microscopy data show that DNA damage, manifested as extensive filamentation and RecA expression, is only prominent during the first generations after persister outgrowth. This suggests that resistance mutations mainly occur during a short time frame, after which these early mutations are being transmitted to future generations.

☐ The authors have checked the abundance of resistance mutations directly after treatment, but it could be interesting to do the same at different time points during the recovery phase. This

experiment could give a rough indication on how long the mutation rate of the persister offspring remains increased.

Figure 1

- Although interesting, these data do not add much to what is reported in Völzing and Brynildsen (2015) about the induction of the SOS response during recovery of persisters and non-persisters. The microscopic observation of outgrowing OFL persisters is new here, but has limited statistical power with only 17 observed persisters, of which 5 are taken into account for quantitative analysis.
- Figure 1A:
 - o It would be interesting to mention the time of first division for all 17 persisters.
 - o As mentioned earlier, it is possible that the results are affected by growth/DNA-duplication or shorter lag of mutant cells during the recovery phase, which obviously would affect the resistance frequencies?
- Figure 1B:
 - o Add a color legend for persister/non-persister
 - o Why are only 5 of the 17 persisters taken into account for quantitative analysis?
 - o 1/5 cells shows weak filamentation and 2/5 cells show weak GFP fluorescence; how representative are the microscopic images in Fig. 1A for the whole persister population?
- Fluoroquinolone-mediated DNA damage is essential in the paper but only demonstrated indirectly by measuring the activity of the SOS-response. It would certainly strengthen the paper if damage was shown in a more direct way under the conditions of the experiment.

Figure 2

- Line 154-155: "...and treated cells failed to fluoresce and elongate to the same extent as wild type cells": this statement should be supported with statistics
- Line 155-157: "Collectively, these data indicate that OFL persisters experience pleiotropic DNA damage and use DNA repair mechanisms during recovery from treatment that enable their survival.": The absence of persisters in the $\Delta recA$ and $lexA3$ mutants shows that the SOS response is required for persister formation, but it does not allow to draw conclusions about the involvement of the SOS response/DNA repair in persister outgrowth. It would be most interesting to have time lapse recordings of $\Delta recA$ or $lexA3$ persisters, to see how these survive antibiotic treatment and what phenotype is displayed (filamentation?), and if this is in agreement with the model.

Figure 3

- Figure 3A:

- o The number of RIF resistant mutants in the untreated population seems to be lower than what was measured under the same conditions in Fig. 3C (wild type) and Fig.3D-F (mutants) (around 10 versus 10-100 resistant CFUs/109 CFUs). Based on the latter, the increase of RIF resistance in the OFL treated population would be considerably smaller. With the OFL data being somewhat misleading due to the enrichment of OFL resistant mutants (as also mentioned by the authors) and the CARB data being non-significant, the overall picture is not very convincing.

- o It is counterintuitive to observe that there is no increase in CARB-resistance in the OFL-derived persisters. Given the model presented in the manuscript, one would expect a difference here as well. Regarding the RIFR mutations, there seems to be less variation in the type of mutations in the RIFR-OFL Persister (position 1576, 1592, 1538) compared to parental and control (although few clones have been analysed). Could it be that some of the mutations have an additional benefit during the recovery phase. This would explain why an increased Rif resistance is seen after recovery and not before, and at the same time explain the Carb data. To make the claims strong here, deep sequencing of many resistant clones or retesting with another antibiotic would be advisable.

- Figure 3C:

- o Multiplying the data before treatment with 100 should give approximately the same results as in Fig. 3A (RIF; untreated), as both represent the abundance of resistant mutants in a stationary phase culture. However, both the mean and variance of the data show quite some difference.

- o The data after treatment are very close to the detection limit of 1 CFU, questioning their reliability (this is not true for the data before treatment, as in this case 100 times as many cells were plated).

- o If these data are correct, they do not support the authors' statement that the recovery phase is necessary for the enhancement of RIF resistance, as the number of resistant mutants per 109 CFUs directly after treatment (Fig. 3C) and after recovery (Fig. 3A) is comparable (i.e. ± 100 resistant CFUs/109 culturable CFUs)

- Figure 3D:

- o It seems that the number of RIF resistant mutants is significantly higher in the untreated population compared to the OFL treated population of the $\Delta recA$ mutant, which is quite surprising as the untreated population did not experience any OFL.

- Figure 3F:

- o When comparing the data to Fig. 3A, the lack of significant difference for $\Delta umuCD$ seems to be caused by an increased resistance development in the untreated population, rather than a decrease in the OFL treated population.

- o Similarly, compared to the data of Fig. 3A, there seems to be an increase in persister levels in all three controls of the $\Delta polB$, $\Delta dinB$ and $\Delta umuCD$ mutants. The relative increase in the $\Delta polB$ and $\Delta dinB$ mutants is also decreased compared to the wt, which would imply their involvement as well.

Each error-prone pol has a mutagenesis signature which could be analysed and make the conclusions more convincing.

Figure 4

- Persister offspring seem to have more resistance mutations than untreated cells, while the general mutation frequency is similar. The authors do not give a clear explanation for this discrepancy.
- A very interesting and new hypothesis is the possible recombination between multiple chromosomes in the elongated cells. Unfortunately, this part is not developed. How would the recombination mediated repair, which is error free, compare to activity of the error prone polymerases? And how does HDR affect resistance development? Both activities seem opposite. Is there any evidence of recombination-mediated repair?

Reviewer #1 (Remarks to the Author):

There are essentially four key points made in the manuscript by Barrett et al. 1) offspring emanating from bacterial persisters (treated with fluoroquinolone) gave rise to more antibiotic-resistant mutants than untreated bacteria; 2) resistance was heritable; 3) resistance depended generally on SOS induction and more specifically on DNA polymerase V; 4) it appears that cells resistant to antibiotics needn't proceed through a tolerance phase described by Balaban et al. (Ref. 8). I believe that ideas and data presented are fresh and important.

We thank the reviewer for his or her careful consideration of our manuscript and positive evaluation of our work.

I have several questions and comments that I'd like the authors to consider.

1. The requirement for pol V in generating antibiotic resistance measured as increased numbers of RIF-resistant colonies emanating from OFL-treated cells (red bars) relative to untreated cells (grey bars), is shown in Fig. 3F. While all three pol mutant strains show a drug-dependent increase in RIF colonies, the differences for Δ umuDC aren't statistically different. Based principally on these small statistical differences, the authors conclude that the increase in mutagenesis of treated cells is dependent specifically on the presence of pol V, and not pols II or IV. This conclusion is reinforced by the restoration of mutagenesis when pol V is overexpressed in a Δ umuDC background (Supp. Fig. 3). Even so, the data are less than overwhelming.

However, let's examine a different comparison. The wild type control data are shown in Fig. 3A, in which the first pair of bars are untreated (dark gray) and OFL treated (red). Additional data are shown in Fig. 3D and Fig. 3E for the recA and lexA3 mutants, respectively. First, compare the data for the untreated cells (dark grey). The wild-type strain yields ~10 RIF colonies per 10^9 CFU. While all the mutant strains yield roughly 30 RIF mutant colonies per 10^9 CFU (Fig 3D and 3E). Why do each of the untreated pol mutant strains give rise to about three times more mutants in the absence of treatment comparing to wild-type untreated strain?

Now let's examine the surviving persisters from the treated cells. The WT (Fig. 3A) yields about 100 RIF colonies per 10^9 CFU on average, which is similar to Δ polB, Δ dinB, and Δ umuDC deletion strains (all yield ~100 RIF colonies per 10^9 CFU). So, yes, if you compare effect for persisters of each mutant strain to untreated mutant control strains, Δ umuDC looks like it might have a reduced effect. However, if instead a comparison is made to treated wild type and treated mutants (which perhaps might be a more relevant way to view the experiment, i.e., same treatment but different strain), there is little if any difference in the number of RIF resistant mutants (compare red bars in Fig. 3A and 3E).

Could the authors comment on this possible alternative interpretation?

At the reviewer's suggestion, we critically thought about the comparisons to perform with our resistant mutant data. For all strains, each fluoroquinolone (FQ)-treated sample is paired to an untreated sample that is processed side-by-side (at the same time with identical media, identical antibiotic-containing plates, etc.). This controlled for any batch-to-batch differences in LB, which is a poorly-defined media. Given the relatively large variations for resistant mutant measurements, we sought to minimize sources of experimental variance (e.g., from LB). For this

reason, we had restricted comparisons to FQ-treated and untreated samples of the same strain because they were always paired, whereas different strains were assayed at different times. To compare between strains, we have elected to use fold-changes of paired samples (FQ-treated/untreated), which is a metric that controls for sources of experimental variance. Those fold-changes are the data that establishes whether an enhancement in resistance development occurs in populations derived from persisters compared to normal cells, and the magnitudes of the fold-changes can be compared between strains to assess the involvement of genetic loci. To illustrate, wild-type cultures exhibited an ~14-fold increase in rifampicin resistance from FQ persisters compared to untreated controls, and that increase was significantly different than the ~4-fold increase observed for $\Delta umuDC$. To facilitate these comparisons by readers, we have modified figures in the main text to portray fold-changes, and the raw data now appears in the Supplementary Information (Supplementary Figures 6A, 7, 8G). In addition, we have elaborated on these methodological details within the manuscript.

Comparing resistance development from untreated samples of different strains will indicate whether significant differences in basal mutation rates are present between strains; whereas analogous comparisons from FQ-treated samples of different strains will quantify the mutation rates in their respective persisters. Neither of those comparisons were the focus of this study, and therefore, side-by-side assays of untreated (or FQ-treated) samples of different strains were not conducted. In the absence of such paired assays, we did not directly compare untreated to untreated and FQ-treated to FQ-treated rates in this study.

2. Regarding the $\Delta recA$ (Fig. 3D) and $lexA3$ (Fig. 3E) data. The $recA$ deletion strain appears to be an antimutator following drug treatment. While there is no asterisk shown, might 8 (drug treated) vs 30 (untreated) colonies reflect an actual difference?

For the $lexA3$ strain, there is virtually no difference in mutants between treated and untreated cells, but again, compared to WT, both have significant reductions compared to wild type in treated cells.

We thank the reviewer for this question. Although the fold-changes observed with these mutants are not significantly lower than 1 ($p=0.16$ for $\Delta recA$ and $p=0.08$ for $lexA3$), it is possible that with additional replicates they may be significantly lower than 1. Given this possibility, we have modified the text indicating that $\Delta recA$ and $lexA3$ may suppress resistance development from persisters.

3. With respect to the induction of the $recA$ -GFP reporter and its relationship to cell length, as shown in Fig. 1B, I see no relationship between GFP induction and filamentation (the proxy for being a persister.) The five persisters are shown in black and, you can see their lengths over time (Fig. 1B, black). I've tried to look at their respective levels of GFP induction. It's not easy to do as presented, but if one looks hard, it's possible to determine which black line in the lower panel corresponds to which black line in the upper panel, because for each cell the data are acquired for different lengths of time. (I strongly suggest that that different shapes and/or colors be used for the five persisters, so that the correlation can be made much more easily.) The problem is that there does not appear to be a correlation in the amount of GFP expression and the time the cell is a persister. If you put the persisters in rank order in length from longest to shortest, 1, 2, ..., 5, and then compare them to the rank order of their GFP levels (i.e., amount of $RecA$ induction), the longest ranks 4th, the next 1st, then 2nd, then 4th, then 3rd. However, an even

more important issue would seem to be that many of the non-persisters (blue lines) appear to show GFP levels greater than the persisters.

We thank the reviewer for this suggestion. We have now modified Fig. 1B such that each of the five persisters are indicated by unique symbols in the length and fluorescence plots. The reviewer is correct that some of the non-persisters are more fluorescent compared with persisters, and we mentioned that fluorescence from the *recA* reporter was not a distinguishing feature between persisters and non-persisters.

4. A general comment on mutation is made in line 241 (“mutations in general, not just those conferring resistance to antibiotics, might be prevalent in the progeny of OFL persisters”). Therefore, the higher the level of general mutation, the better the chance for increased antibiotic resistance. The mutation frequency for RIF is $10e-7$, which would seem to be in the range of typical mutation frequencies measured in batch culture. Presumably, the authors would attribute this seeming equivalence to the accurate repair of most DNA polymerase-catalyzed misincorporations. Alternatively, however, might it be reasonable to suggest that there is perhaps no transient increase in mutation frequency, but rather the level of mutations might instead be the baseline frequency expected for a phenotype having a large base pair target size?

We are not sure we fully understood the reviewer’s question; however, in this revision we demonstrate that OFL persisters enhance resistance development to rifampicin, carbenicillin, fosfomycin, and D-cycloserine. All of those antibiotics target different enzymes or complexes, and resistance is conferred by different mutations, which do not necessarily have the same base pair target size.

5. In several places in the manuscript, the authors refer to “extensive” or “massive” amount of DNA damage the cells have experienced. However, the amount is not quantitated, and I don’t think they actually do try to link it to GFP expression directly. However, the authors do imply that if cells are filamenting, then there must be a large amount of DNA damage. Is that actually true? Wouldn’t a cell with a single DSB continue to filament until the damage is repaired? And, if one-third of the cells only have one chromosome, and the authors suggest that two chromosomes are needed for homologous recombination-based repair, then perhaps those cells having a single chromosome might filament for a long time. I suggest that the authors comment on this point.

We thank the reviewer for this comment. As we did not quantify the amount of DNA damage in cells following OFL treatment, we have removed adjectives such as “extensive” and “massive” in our text. Based on our time-lapse microscopy data, we infer that the cells have experienced DNA damage based on their induction of the SOS response and filamentation, both of which are classic signs of DNA damage. In this revision, we have stained untreated and OFL-treated wild type cultures with DAPI, a nucleic acid stain, and show that treated cultures show abnormal nucleoid morphology (Supplementary Fig. 2). We note that this method has been used by other groups to demonstrate DNA damage (Sharma *et al.*, 2017, *Nature Comm.* **8**: 1444; Keyamura *et al.*, 2013, *J. Biol. Chem.* **288**:29229-29237; Chai *et al.*, 2014, *J. Biol. Chem.* **289**: 11342-11352). Specifically, we demonstrate that the nucleoid morphology of OFL-treated cells resembles that of cells treated with mitomycin C, a DNA cross-linking agent that results in DNA double-stranded breaks and filamentation, whereas their nucleoid morphology is distinct from that of

cells treated with piperacillin, which causes filamentation in the absence of DNA damage by inhibiting PBP3. This new data provides further evidence that OFL-treated cells experience DNA damage; however it does not quantify the number of breaks and we now mention in the discussion that the abundance, type, and residence time of DNA lesions are questions for future studies.

6. Suggestions concerning the references:

a. For the experiment where the number of chromosomes per cell are determined (line 264), the work of Bernander and colleagues (Supplement Ref. 19) should be cited in the text references. In Supplement Ref. 19, Akerlund et al. showed that stationary phase E. coli can have 1, 2, 4 or 8 chromosomes. (Barrett et al., report 1, 2, 3, and 4.) The authors should explain where cells with 3 chromosomes come from. Generally, it is thought that if the origins are going to fire multiple times, they fire together. At least that appears to follow from Akerlund et al., since they did not observe 3 or 5 or 6 chromosomes, just 1, 2 and powers of 2.

Following this suggestion, we have included a reference for the paper by Bernander and colleagues in the main text in addition to the Supplementary Information. As for the chromosome number quantification, we note that ~97% of the cells in our stationary phase conditions have 1, 2, or 4 chromosomes. Using the method of Akerlund and colleagues, our flow cytometry data indicate that 50,000 fluorescent units correspond to cells with one chromosome. By that metric, the small peak around 150,000 fluorescent units should correspond to cells with three chromosomes. In the study by Akerlund and colleagues, a minor bulge in the flow cytometry profile that corresponds to three chromosomes is present in Figure 4, and using their technique we also observed a small peak at approximately three chromosomes in long-term LB-glucose stationary phase cultures. Notably, Nielsen and colleagues (Nielsen *et al.*, 2006, *Mol. Microbiol.* **61**: 383-393) demonstrated using a fluorescent origin reporter that in a population of cells with inhibited replication and cell division, a small fraction of cells (~2%) contained three chromosomes. It is possible that interruptions in replication and division contributed to the presence of cells with three chromosomes in stationary phase cultures; however, we did not delve more deeply into confirming or refuting that hypothesis. In the revised manuscript, we mention the consistency between our data and the previous studies where *E. coli* has been observed with three chromosomes.

b. Reference 31 for the encoding of pol II should be replaced by the primary citation: Bonner, C.A., Hays, S., McEntee, K., and Goodman, M.F. DNA Polymerase II is Encoded by the DNA Damage-Inducible dinA Gene of Escherichia coli. Proc. Natl. Acad. Sci. USA 87, 7663-7667 (1990).

We thank this reviewer for his or her advice and we have replaced reference 31 with the one by Bonner and coworkers.

c. Reference 33 for the encoding of pol V should be replaced by the primary citation: Tang, M., Shen, X., Frank, E. G., O'Donnell, M., Woodgate, R., and Goodman, M. F. UmuD'2C is an error-prone DNA polymerase, Escherichia coli pol V. Proc. Natl. Acad. Sci. USA 96, 8919-8924 (1999).

We thank the reviewer for this suggestion and we have replaced reference 33 with the one by Tan and coworkers.

d. On line 222 the authors refer to the “three nonessential polymerases.” But these three pols are non-essential in monoculture. They are essential for long-term survival in co-culture as shown in Yeiser, B., Pepper, E. D., Goodman, M. F., and Finkel, S. E. SOS-induced DNA polymerases enhance long-term survival and evolutionary fitness. Proc. Natl. Acad. Sci. USA 99, 8737-8741 (2002). As a closely related point: the authors say that the three pols “participate in DNA synthesis in response to DNA damage” (Line 223-224). That’s correct, but the three pols also participate in DNA synthesis in the absence of exogenous DNA damaging agents as shown in Corzett, C. H., Goodman, M. F. and Finkel, S. E. Competitive Fitness during Feast and Famine: How SOS DNA Polymerases Influence Physiology and Evolution in Echerichia coli Genetics 194, 409-420 (2013).

In response to these suggestions, we modified our sentence and replaced “three nonessential polymerases” with “three SOS-inducible polymerases.” Further, we added a statement indicating that the error-prone polymerases participate in long-term stationary phase adaptation, citing the work of Yeiser and colleagues, and also referenced the paper by Corzett and coworkers.

Reviewer #2 (Remarks to the Author):

This manuscript by Barret et al examines how a single exposure of persister cells with fluoroquinolones affect the frequency of resistant mutants using time-lapse microscopy, mutation frequency measurements and genetic analysis. The authors show that FQ induces the SOS response and suggest that this results in a strong increase in mutation frequency in the recovered persister population.

The most important experiments in this manuscript are the measures of mutation frequencies. Unfortunately, they are not fully described and they also lack some key controls that I think are needed before one can conclude what the authors suggest is happening (i.e. 100-fold increase in mutation frequency in persisters in response to FQ exposure)

Major points

1. The mutation frequency assay is not as clearly described as it should be. Key information is lacking regarding how many cells were inoculated into what volume from the fluoroquinolone and untreated control? Thus, to make a fair comparison of mutation frequencies between treated and untreated populations one needs to know the number of generations of growth for each population (see also point 2 below).

We thank the reviewer for his or her careful consideration of our manuscript and constructive suggestions. Following this suggestion, we performed additional experiments where we quantified the number of colony forming units (CFU) of the untreated and OFL-treated population inoculated into LB at the onset of the post-treatment recovery period (Supplementary Fig. 6D). We further monitored the change in OD₆₀₀ and CFU/mL at 2, 4, 8, and 16 h of the recovery period (Supplementary Fig. 6C and 6D). We determined that we inoculated $8.7 \times 10^7 \pm 1.4 \times 10^7$ CFU/mL of the untreated population, whereas $4.8 \times 10^6 \pm 6.5 \times 10^5$ CFU/mL of the OFL-treated population was inoculated. By 2 h of recovery, the population of OFL-treated cells decreased to $1.8 \times 10^6 \pm 3.3 \times 10^5$ CFU/mL and then increased continuously. Based on these observations, we diluted the untreated inoculum 50- and 100-fold to more closely match the CFU counts of the OFL-treated cells and examined the effect of these dilutions on resistance toward rifampicin (RIF). We found that the number of RIF resistant colonies originating from cultures derived from the OFL persisters were significantly higher than the untreated cultures, 50-fold diluted untreated cultures, and 100-fold diluted untreated cultures (Supplementary Fig. 6E).

2. Another important point regarding the mutation frequency measurement is to what extent during the recovery phase, the death rate is higher in the treated population as compared to the non-treated. Thus, if there is a higher death rate in the treated population (for example due to the accumulated DNA damage or presence of intracellular residual FQ) one will underestimate the number of generations of growth and the mutation frequency will appear higher (even though there is in fact no increase in the intrinsic mutation rate). A recent and very important paper from the Bonhoeffer laboratory shows that this generally results in overestimation of mutation rates during antibiotic exposure. Thus, I would strongly suggest that the authors measure the number of divisions with the plasmid system used in Frenoy and Bonhoeffer (PLoS Biol. 2018, 16(5):e2005056) to examine/exclude this possibility.

We thank the reviewer for this suggestion. We considered using the plasmid system described in Frenoy and Bonhoeffer (*PLoS Biol.*, 2018), but realized that their methodology involves assumptions that do not apply to our experimental conditions. For example, their system assumes that each cell is monoploid; however, in the populations we studied, only ~30% of the population had one chromosome at the beginning of the growth period preceding the assessment of resistant mutants (Fig. 4B), and in that growth media (LB) only a minority of *E. coli* have one chromosome with some having up to eight (Akerlund *et al.*, 1995, *J. Bacteriol.* **177**: 6791-6797). Further, their model assumes that the population exhibits homogenous behavior with characteristic division and death rates in the presence of antibiotics; however, our study focuses on phenotypic heterogeneity, and specifically, cells that differ in their outcomes following antibiotic treatment. The essence of the Frenoy and Bonhoeffer study was that for comparison of mutation rates, the number of times the genome has been replicated under different conditions needed to be accounted for. When we monitored OD₆₀₀ and CFUs/mL of both the untreated and OFL-treated cultures, we observed that while the OD₆₀₀ of both cultures tracked similarly and ended up with a similar terminal OD₆₀₀, CFUs/mL differed except for the final time point (Supplementary Fig. 6C and D). Since the number of CFUs at t=0 recovery in the OFL-treated population were about 50- to 100-fold less than that of the untreated control, we conducted control experiments, whereby the untreated control was diluted an additional 50- or 100-fold into LB for recovery. Our data indicated that those additional generations of growth do not explain the increase in RIF resistant colonies that arise from OFL persisters (Supplementary Fig. 6E). Further, we sought to provide a direct measure of genome amplification, and therefore, quantified the total DNA abundance at the beginning and end of recovery and outgrowth experiments. As depicted in Supplementary Fig. 6F, the untreated and OFL persister populations amplified their DNA to comparable magnitudes. We note that the culturability measurements for the OFL persister samples initially dropped prior to expanding exponentially, and that some DNA could have been released into the extracellular space. We confirmed that any extracellular DNA was stable for 16 hours (Supplementary Fig. 6G), and thus contributed to the DNA amplification measurements performed on untreated and OFL persister samples.

3. It is interesting to note that in other studies, LexA defective mutants with constitutive SOS induction show a very moderate (a few-fold) increase in mutation rates. How does that square with the present observation of a 100-fold increase? Does it not suggest that there are other factors than the SOS induction per se (and an associated increase in PolV) that cause the apparent increase in mutation frequency? It would be interesting to see if artificial over-expression of PolV (to the same level as that seen during SOS induction) results in the same level of mutagenesis.

We thank the reviewer for his or her comment. We would like to clarify that upon OFL treatment and recovery, we observed a ~14-fold increase in RIF resistance compared with the untreated control. The ~100-fold increase was for OFL-resistant mutants, and we did not perform resistance development experiments with that antibiotic and *lexA3* due to the potential contributions from pre-existing OFL-resistant mutants prior to the persistence assay. During the course of revising this manuscript, we examined resistance enhancement toward additional antibiotics, and we observed fold-enhancements that ranged from 6 to 20-fold (excluding that for OFL resistant mutants). In studies conducted with *lexA*-defective (*lexA51*) mutants (Ennis *et al.*,

1985, *PNAS* **82**: 3325-3329; Watanabe-Akanuma *et al.*, 1997, *Mutat. Res.* **373**: 61-66), increases in mutagenesis were not observed in the absence of DNA damage unless the strain carried both the *lexA51* allele and the *recA730* allele, which encodes a RecA mutant with constitutively active co-protease functions. In strains harboring *lexA51* and *recA730*, Ennis and colleagues reported ~13-fold increase in mutagenesis in the absence exogenous DNA damage compared with *lexA+recA+* strains. Similarly, Watanabe-Akanuma and colleagues reported ~15 to 39-fold (depending on the nature of transversion mutation examined) increases in mutagenesis in cells with *lexA51* and *recA730* mutations compared with wild-type cells (~15-fold for A:T→C:G transversions, ~17-fold for G:C→T:A transversions, and ~39-fold for A:T→T:A transversions). Thus, the fold-increase in resistant mutants that we observed following recovery from FQ treatment are comparable to those reported in studies conducted with defective LexA and RecA with constitutive co-protease activity. Likewise, it was previously reported that an *E. coli* strain that constitutively expressed *umuDC* exhibited a ~10-fold increase in mutagenesis in the absence of DNA damage (Sommer *et al.*, 1993, *Mol. Gen. Genet.* **239**:137-144), which is also comparable to the fold-changes we observed.

4. One of the authors' main points is that persisters might contribute to evolution of acquired resistance during treatment. But to be fair in that analysis should one not take into account the relative contributions from the non-persisters and persisters with regard to mutant formation? That is, the likelihood that resistant mutants will appear in the respective populations is: fraction of total cells x relative mutation rate. For normal cells this would: 0.9999 x 1 = 0.9999 and for persisters: 0.0001 (assuming that the persister fraction is 10E-4) x 100 = 0.01, indicating that even with a 100-fold increase in mutation rate in the persister population the large majority of mutants are still derived from the normal population. I think this ought to be explicitly discussed.

We thank the reviewer for his or her comment. In this manuscript, we investigate stationary phase populations where OFL persisters comprise approximately 2% of the population (Fig. 2B, Supplementary Fig. 1A). We chose to study growth-inhibited populations because they are the most difficult to eradicate clinically. For efficacious OFL treatments, which leave persisters as the only surviving cell type, all resistant mutants arising from the subsequent population would originate from persisters. A main message of this article is that FQ persisters do not conform to the paradigm where antibiotics fail to corrupt their primary target and thus resistance development resembles what would be expected for normal cell populations. Rather, resistant mutants arise far more frequently from FQ persisters than normal cells. Thus the contribution from FQ persistence would be a comparison of resistant mutants from a normal population (untreated) and that derived from FQ persisters (following FQ treatment), which for RIF is ~14-times more resistant mutants on average from the FQ persister-derived population compared to the untreated population. Thus the vast majority of RIF-resistant cells present after an efficacious FQ treatment would be because the population arose from FQ persisters. We have clarified these points in the manuscript.

Other comments

5. Line 33. What is actually the evidence that they are not genetically different?

The statement is based on our whole genome sequencing data (depicted in Figure 4A and S4A-C) of 4 colonies from parental cultures, 10 colonies from untreated populations, and 10 colonies from OFL persisters. Our analysis of the sequencing data did not detect any genetic differences from those 14 colonies.

6. Line 34. "Favorable phenotypic niche? Sounds nice but what does it mean?"

We thank the reviewer for asking this question. In response to this comment, we have clarified the text to explain that "favorable phenotypic niche" refers to a molecular physiology with regard to the abundances of DNA, RNA, proteins, metabolites, and other cellular components that increase the tolerance of individual bacteria to specific antibiotics.

Reviewer #3 (Remarks to the Author):

This is a very elegant study showing that fluoroquinolone persisters develop a strong SOS response leading to the formation of antibiotic-resistant mutants. This effect is mediated by RecA and mutagenic polymerases. The manuscript is straightforward and logically structured. This work is novel and may have important consequences for the development of antibiotic-resistance in a clinical context.

We thank the reviewer for his or her careful consideration of our manuscript and positive evaluation of our work.

Please find below my detailed comments.

General remarks

- Despite the washing steps performed after OFL treatment, low OFL concentrations (i.e. 0.002 µg/ml or 1/32x MIC) remain present in the samples. A control experiment should be performed where untreated cells are put on an agar pad containing 0.002 µg/ml OFL, to confirm that this concentration does not induce the observed phenotypic effects of filamentation and SOS response activation. Similarly, an experiment could be performed where untreated cells are put in LB containing 0.002 µg/ml OFL for 16h and plated onto RIF plates, to exclude the potential effect of subinhibitory OFL concentrations on the mutation rate (as reported by Kohanski et al., 2010).

We thank the reviewer for these suggestions. We have conducted time-lapse microscopy of untreated cells plated on an agar pad containing trace OFL. In our experiments, we washed 1 mL of cells treated with 5 µg/mL of OFL twice with 1 mL PBS, thereby reducing the concentration of the antibiotic by ~2,500-fold and resulting in ~0.002 µg/mL of OFL. For microscopy experiments, we further diluted the cells 12-fold with spent media derived from untreated cultures, resulting in ~0.2 ng/mL OFL. Thus, we conducted our time-lapse microscopy experiments with untreated cells plated on agar pads with 0.2 ng/mL OFL. We observed that these cells did not filament or induce the SOS response (Supplementary Video 4; Supplementary Fig. 3C). We also recovered untreated cells in the presence of trace OFL and examined the impact of trace antibiotics on enhancement of RIF resistance. In our assays, we washed OFL-treated cells twice with 1 mL PBS and concentrated the sample to 300 µL before inoculating this 300 µL culture in 25 mL of LB. This dilutes the concentration of OFL to ~0.02 ng/mL. As such, we cultured untreated populations in LB with 0.02 ng/mL OFL for 16 h before plating 10⁹ cells derived from this population on LB agar with 500 µg/mL RIF, and we did not observe resistance enhancement above untreated cells cultured in LB without OFL (Supplementary Fig. 6B).

- The authors claim that the 16h recovery phase is required for the enhancement of RIF resistance. However, the microscopy data show that DNA damage, manifested as extensive filamentation and RecA expression, is only prominent during the first generations after persister outgrowth. This suggests that resistance mutations mainly occur during a short time frame, after which these early mutations are being transmitted to future generations.

The authors have checked the abundance of resistance mutations directly after treatment, but it could be interesting to do the same at different time points during the recovery phase. This experiment could give a rough indication on how long the mutation rate of the persister offspring remains increased.

We thank the reviewer for his or her suggestion to perform a time course analysis of resistance development during recovery, and have included our data in Fig. 2D. We measured resistance development at 8, 12, and 16 hours of recovery, and initially attempted to perform analogous measurements at 2 and 4 hours. Unfortunately, at 2 and 4 hours we could not harvest 10^9 cells from persister-derived cultures. Notably, for all time points considered, RIF resistance was enhanced compared to untreated controls, which suggested that the mutations arise prior to the 8 hour time point.

Figure 1

- Although interesting, these data do not add much to what is reported in Völzing and Brynildsen (2015) about the induction of the SOS response during recovery of persisters and non-persisters. The microscopic observation of outgrowing OFL persisters is new here, but has limited statistical power with only 17 observed persisters, of which 5 are taken into account for quantitative analysis.

We agree with the reviewer that the results are consistent with those of Volzing and Brynildsen; however, we believe that they do provide further knowledge of how FQ persisters recover from treatment, because they illustrate the extent of filamentation, which was not quantified by Volzing and Brynildsen, and they provide much better time resolution to the recovery process.

- Figure 1A:

o It would be interesting to mention the time of first division for all 17 persisters.

While we observed 17 persisters overall, we were only able to quantify the length and fluorescence of five persisters as the other 12 persisters were incompletely imaged; they either exited the frame as they lengthened or they entered the frame during the experiment (*i.e.*, they were not observed at the beginning of the time-lapse). Divisions of these cells were observed, as well as a return to normal shape, size, and lack of SOS induction in their progeny in order to be identified as persisters; however, with only part of the cell within the visible field, it was not possible to ascertain the time of first division, because that specific event could have been in an un-visualized portion of the cell.

o As mentioned earlier, it is possible that the results are affected by growth/DNA-duplication or shorter lag of mutant cells during the recovery phase, which obviously would affect the resistance frequencies?

We thank the reviewer for this question. We performed experiments with untreated and OFL-persister samples where the starting CFU/mL levels were comparable, and significant resistance enhancement was still observed for the populations derived from OFL persisters (Supplementary Fig. 6). We also quantified DNA abundances in untreated and OFL persister-derived cultures and observed similar levels of DNA amplification. In addition, if RIF-resistant persisters took less time to recover from OFL treatment than non-RIF resistant persisters, or if RIF-resistant mutants from OFL persisters exhibited faster growth rates than RIF-resistant mutants from untreated populations, we reason that their relative abundances would increase, and that both of those possibilities would still represent an enhancement in resistance development from OFL

persisters. We now mention these ways in which OFL persisters could potentially enhance resistance development in the main text.

- *Figure 1B:*

o Add a color legend for persister/non-persister

We thank the reviewer for this suggestion and have added a color legend for persisters and non-persisters in this Figure.

o Why are only 5 of the 17 persisters taken into account for quantitative analysis?

As noted above, we observed 17 persisters overall, but we only quantified the length and fluorescence of five persisters because the other 12 persisters either exited the frame as they lengthened or they entered the frame during the experiment (*i.e.*, they were not observed at the beginning of the time-lapse).

o 1/5 cells shows weak filamentation and 2/5 cells show weak GFP fluorescence; how representative are the microscopic images in Fig. 1A for the whole persister population?

We thank the reviewer for this question. To demonstrate that the persister shown in Fig. 1A is representative of the 17 persisters we observed from our time-lapse microscopy experiments, we now show the brightfield and fluorescent images of the 12 persisters that were not subjected to quantitative analysis (Supplementary Fig. 4). Those images precede the observed cell division event that indicated that those cells were persisters, although as noted above it may not have been the first division event. We observed that the majority of the 17 persisters filamented and were fluorescent.

- Fluoroquinolone-mediated DNA damage is essential in the paper but only demonstrated indirectly by measuring the activity of the SOS-response. It would certainly strengthen the paper if damage was shown in a more direct way under the conditions of the experiment.

We thank the reviewer for his or her suggestion. In response, we direct the reviewer to our previous work (Volzing and Brynildsen, 2015, *mBio* **6**: e00731-15), which demonstrates that DNA repair machinery (such as *recA* and *recB*) are critical for persistence to OFL in stationary phase cultures, and we also have included new data implicating homologous recombination in the OFL persistence phenotype by demonstrating decreased persister levels for mutants that lack the ability to resolve Holliday junctions (Δ *ruvA* Δ *recG*), which are comparable to that of a Δ *recB* mutant. We have also now stained untreated and OFL-treated wild type cultures with DAPI, a nucleic acid stain, and show that treated cultures show abnormal nucleoid morphology (Supplementary Fig. 2). We note that this method has been used by other groups to demonstrate DNA damage (Sharma *et al.*, 2017, *Nature Comm.* **8**: 1444; Keyamura *et al.*, 2013, *J. Biol. Chem.* **288**:29229-29237; Chai *et al.*, 2014, *J. Biol. Chem.* **289**: 11342-11352). Specifically, we demonstrate that the nucleoid morphology of OFL-treated cells resembles that of cells treated with mitomycin C, a DNA cross-linking agent that results in DNA double-stranded breaks and filamentation, whereas their nucleoid morphology is distinct from that of cells treated with

piperacillin, which causes filamentation in the absence of DNA damage by inhibiting PBP3. This new data provides further evidence that OFL-treated cells experience DNA damage.

Figure 2

- Line 154-155: "...and treated cells failed to fluoresce and elongate to the same extent as wild type cells": this statement should be supported with statistics

Following the reviewer's suggestions, we have included lines in Fig. 1B, 1C, Supplementary Fig. 3C, 5C, 5D, 5G, and 5H to indicate the cell length and fluorescence values that contain 95% of the measured points for the indicated strains and conditions. We note that 26.4% of the OFL-treated wild-type cell length data lies above the 95% interval for the OFL-treated $\Delta recA$, and 19.6% lie above the 95% interval for $lexA3$ mutants. With respect to fluorescence, 88.6% of the OFL-treated wild-type data lies above the 95% interval for the OFL-treated $\Delta recA$, and 82.7% lie above the 95% interval for $lexA3$ mutants.

- Line 155-157: "Collectively, these data indicate that OFL persisters experience pleiotropic DNA damage and use DNA repair mechanisms during recovery from treatment that enable their survival.": The absence of persisters in the $\Delta recA$ and $lexA3$ mutants shows that the SOS response is required for persister formation, but it does not allow to draw conclusions about the involvement of the SOS response/DNA repair in persister outgrowth. It would be most interesting to have time lapse recordings of $\Delta recA$ or $lexA3$ persisters, to see how these survive antibiotic treatment and what phenotype is displayed (filamentation?), and if this is in agreement with the model.

Following the reviewer's recommendation, we have included time-lapse recordings of the $\Delta recA$ and $lexA3$ persisters, which demonstrate that these mutants do not filament or fluoresce appreciably after OFL treatment. We note that we did not observe any persisters arising from these mutants using microscopy, which was to be expected given the very low frequency of persisters in these mutants ($2.8 \times 10^{-4} \pm 9.64 \times 10^{-5}$ for $\Delta recA$ and $1.4 \times 10^{-5} \pm 2.1 \times 10^{-6}$ for $lexA3$).

Figure 3

- Figure 3A:

o The number of RIF resistant mutants in the untreated population seems to be lower than what was measured under the same conditions in Fig. 3C (wild type) and Fig.3D-F (mutants) (around 10 versus 10-100 resistant CFUs/109 CFUs). Based on the latter, the increase of RIF resistance in the OFL treated population would be considerably smaller. With the OFL data being somewhat misleading due to the enrichment of OFL resistant mutants (as also mentioned by the authors) and the CARB data being non-significant, the overall picture is not very convincing.

We appreciate the reviewer's comment. For all strains, each FQ-treated sample was paired to an untreated sample that was processed side-by-side (at the same time with identical media, identical antibiotic-containing plates, etc.). This controlled for any batch-to-batch differences in LB, which is a poorly-defined media. Given the relatively large variations for resistant mutant measurements, we sought to minimize sources of experimental variance (e.g., from LB). For this reason, we previously restricted comparisons to FQ-treated and untreated samples of the same strain because they were always paired, whereas different strains were assayed at different times.

The data commented on here by the reviewer were obtained at different times, and thus not directly compared in the original manuscript. To compare between strains, we have now elected to use fold-changes of paired samples (FQ-treated/untreated), which is a metric that controls for sources of experimental variance. Those fold-changes are the data that establishes whether an enhancement in resistance development occurs in populations derived from persisters compared to normal cells, and the magnitudes of the fold-changes can be compared between strains to assess the involvement of genetic loci. For RIF, FQ persister-derived populations yielded ~14-fold more resistant mutants on average than untreated populations. To provide additional evidence that FQ-persister derived populations accelerate resistance development, we performed additional replicates for CARB and performed new assays with fosfomycin and D-cycloserine. FQ persister-derived populations for all three produced more antibiotic resistant mutants than untreated controls (CARB: ~6-fold, FOS: ~6-fold, DCS: ~20-fold). We believe that these additional experiments have greatly increased the potential implications of our study. We also note that while the fold-change data is presented in the main text, the raw data is presented in the Supplementary Information (Supplementary Figures 6A, 7, 8G), and we explain in the methods why fold-change data was used for comparisons.

o It is counterintuitive to observe that there is no increase in CARB-resistance in the OFL-derived persisters. Given the model presented in the manuscript, one would expect a difference here as well.

We thank the reviewer for his or her comment, and conducted additional replicates of our assay, resulting in a statistically significant increase in CARB-resistance from OFL persisters.

Regarding the RIFR mutations, there seems to be less variation in the type of mutations in the RIFR-OFL Persister (position 1576, 1592, 1538) compared to parental and control (although few clones have been analysed). Could it be that some of the mutations have an additional benefit during the recovery phase. This would explain why an increased Rif resistance is seen after recovery and not before, and at the same time explain the Carb data. To make the claims strong here, deep sequencing of many resistant clones or retesting with another antibiotic would be advisable.

At the reviewer's suggestion, we tested additional antibiotics and performed additional replicates of CARB. For all antibiotics tested, OFL-persister derived populations led to more resistant mutants than untreated controls.

- Figure 3C:

o Multiplying the data before treatment with 100 should give approximately the same results as in Fig. 3A (RIF; untreated), as both represent the abundance of resistant mutants in a stationary phase culture. However, both the mean and variance of the data show quite some difference.

o The data after treatment are very close to the detection limit of 1 CFU, questioning their reliability (this is not true for the data before treatment, as in this case 100 times as many cells were plated).

o If these data are correct, they do not support the authors' statement that the recovery phase is necessary for the enhancement of RIF resistance, as the number of resistant mutants per 109 CFUs directly after treatment (Fig. 3C) and after recovery (Fig. 3A) is comparable (i.e. ± 100 resistant CFUs/109 culturable CFUs)

As described above, previously we did not directly compare the resistant mutant levels from these experiments, because with and without recovery experiments were not conducted in parallel. However, the fold-changes in resistance development from paired samples can be compared, because that metric controls for experimental variation. In addition, we repeated the without recovery experiment and plated more cells (10 mL of cultures on five LB agar plates with RIF as opposed to plating 500 μ L of the culture on one plate as we had done initially) in order to improve the limit of detection. Fold-change data from those experiments demonstrate that post-treatment recovery is indeed needed for enhancement of RIF resistance (Fig. 2C).

- Figure 3D:

o It seems that the number of RIF resistant mutants is significantly higher in the untreated population compared to the OFL treated population of the Δ recA mutant, which is quite surprising as the untreated population did not experience any OFL.

We thank the reviewer for this comment. While the fold change of RIF resistant colonies arising from FQ persister-derived and untreated populations of Δ recA were not statistically lower than 1 at a significance threshold of $p \leq 0.05$, we recognize that it was close to being significant ($p = 0.16$). Thus, we have modified our text to mention the possibility that *recA* deletions might suppress resistance development from persisters.

- Figure 3F:

o When comparing the data to Fig. 3A, the lack of significant difference for Δ umuCD seems to be caused by an increased resistance development in the untreated population, rather than a decrease in the OFL treated population.

o Similarly, compared to the data of Fig. 3A, there seems to be an increase in persister levels in all three controls of the Δ polB, Δ dinB and Δ umuCD mutants. The relative increase in the Δ polB and Δ dinB mutants is also decreased compared to the wt, which would imply their involvement as well. Each error-prone pol has a mutagenesis signature which could be analysed and make the conclusions more convincing.

We now present the data as fold-change in resistant mutant levels in FQ-persister and untreated populations for the reasons described above. Using this metric, the fold-change in RIF resistant mutant enhancement for Δ dinB and Δ polB are indistinguishable from wild-type, whereas Δ umuDC is significantly lower.

Figure 4

- Persister offspring seem to have more resistance mutations than untreated cells, while the general mutation frequency is similar. The authors do not give a clear explanation for this discrepancy.

We appreciate the reviewer's question. Resistance mutations are rare events, 1 in 1-10 million cells, and for the whole genome sequencing data that we obtained (10 colonies of treated, 10 colonies of untreated) it is reasonable to expect that none of them would harbor a resistance mutation. Therefore, it is not a discrepancy, but rather a reflection of the relative abundance of resistant mutants.

- A very interesting and new hypothesis is the possible recombination between multiple chromosomes in the elongated cells. Unfortunately, this part is not developed. How would the recombination mediated repair, which is error free, compare to activity of the error prone polymerases? And how does HDR affect resistance development? Both activities seem opposite. Is there any evidence of recombination-mediated repair?

Following this reviewer's suggestion, we have included new data implicating homologous recombination in the FQ persistence phenotype by demonstrating decreased survival in mutants that lack the ability to resolve Holliday junctions ($\Delta ruvA\Delta recG$), which is comparable to that of mutants that cannot process double stranded breaks ($\Delta recB$). We also direct the reviewer to Volzing and Brynildsen (2015), which provides further evidence of recombination-mediated repair in OFL persists from stationary phase populations. Previous studies have demonstrated that following DNA damage, RecA and genes involved in homologous recombination, which repair lesions with high fidelity, are induced well before the induction of error-prone polymerases (reviewed in Michel, 2005, *PLoS Biol.* **3**:e255). Thus, if the repertoire of homologous recombination enzymes fail to fully repair lesions, error-prone polymerases are activated as a last-ditch effort for repair and elevated levels of these mutagenic enzymes in individual cells can facilitate the development of mutations leading to resistance.

REVIEWERS' COMMENTS:

Reviewer #1 (Remarks to the Author):

The authors have responded appropriately to our comments, criticisms, and suggestions.

Myron F. Goodman

Reviewer #2 (Remarks to the Author):

none

Reviewer #3 (Remarks to the Author):

none

REVIEWERS' COMMENTS:

Reviewer #1 (Remarks to the Author):

The authors have responded appropriately to our comments, criticisms, and suggestions.

Myron F. Goodman

Reviewer #2 (Remarks to the Author):

none

Reviewer #3 (Remarks to the Author):

none

We thank the reviewers for their careful consideration of our manuscript, constructive suggestions, and positive evaluation of our work.